# miR-9 modulates and predicts the response to radiotherapy and EGFR inhibition in HNSCC

Francesca Citron[1,2] (ID), Ilenia Segatto[1], Lorena Musco[1], Ilenia Pellarin[1], Gian Luca Rampioni Vinciguerra[1,3], Giovanni Franchin[4], Giuseppe Fanetti[4], Francesco Miccichè[5], Vittorio Giacomarra[6], Valentina Lupato[6], Andrea Favero[1], Isabella Concina[1], Sanjana Srinivasan[2], Michele Avanzo[7] (ID), Isabella Castiglioni[8,9], Luigi Barzan[6], Sandro Sulfaro[10], Gianluigi Petrone[5,†], Andrea Viale[2], Giulio F Draetta[2], Andrea Vecchione[3], Barbara Belletti[1] (ID) & Gustavo Baldassarre[1,*] (ID)

## Abstract

Radiotherapy (RT) *plus* the anti-EGFR monoclonal antibody Cetuximab (CTX) is an effective combination therapy for a subset of head and neck squamous cell carcinoma (HNSCC) patients. However, predictive markers of efficacy are missing, resulting in many patients treated with disappointing results and unnecessary toxicities. Here, we report that activation of EGFR upregulates miR-9 expression, which sustains the aggressiveness of HNSCC cells and protects from RT-induced cell death. Mechanistically, by targeting KLF5, miR-9 regulates the expression of the transcription factor Sp1 that, in turn, stimulates tumor growth and confers resistance to RT+CTX *in vitro* and *in vivo*. Intriguingly, high miR-9 levels have no effect on the sensitivity of HNSCC cells to cisplatin. In primary HNSCC, miR-9 expression correlated with Sp1 mRNA levels and high miR-9 expression predicted poor prognosis in patients treated with RT+CTX. Overall, we have discovered a new signaling axis linking EGFR activation to Sp1 expression that dictates the response to combination treatments in HNSCC. We propose that miR-9 may represent a valuable biomarker to select which HNSCC patients might benefit from RT+CTX therapy.

**Keywords** EGFR inhibitors; HNSCC; KLF5; radiotherapy; Sp1
**Subject Categories** Biomarkers; Cancer

## Introduction

Head and neck squamous cell carcinomas (HNSCC) comprehend a group of epithelial malignancies of the oral cavity, oropharynx, larynx, or hypopharynx with more than 890,000 new cancer cases and 450,000 deaths/year worldwide (Bray *et al*, 2018; Siegel *et al*, 2018). More than 60% of patients are diagnosed with locally advanced disease (stage III/IV), and, despite improvement in multimodal personalized therapies, 50% of these patients will eventually experience recurrence (Adelstein *et al*, 2003; Haddad & Shin, 2008). Standard of care for these patients, who are not eligible for curative surgery, is either concurrent definitive chemoradiotherapy or chemotherapy alone, with the former approach appearing to be the most promising (Bar-Ad *et al*, 2014). Chemotherapy consists of a platinum-based regimen, usually cisplatin (CDDP), or carboplatin for less fit patients, or a combinatorial chemotherapy, including a taxane (paclitaxel) or 5-fluorouracil (5-FU) (Chow, 2020). Since these options are associated with similar survival benefits (Bar-Ad *et al*, 2014), single-agent CDDP is the most commonly used treatment in combination with radiotherapy (RT) (Bar-Ad *et al*, 2014).

Cetuximab (CTX), an EGFR-targeting monoclonal antibody, was the first targeted therapy that received FDA approval in combination with RT to treat locally or regionally advanced HNSCC (Santuray *et al*, 2018), showing reduced toxicity compared with standard RT+CDDP. Interestingly, a recent meta-analysis on 4,212 patients treated with RT+CDDP versus RT+CTX showed no differences between the two groups when 3-year survival and recurrence were considered (Huang *et al*, 2016).

1   Molecular Oncology Unit, Centro di Riferimento Oncologico di Aviano (CRO), IRCCS, National Cancer Institute, Aviano, Italy
2   Department of Genomic Medicine, The University of Texas MD Anderson Cancer Center, Houston, TX, USA
3   Faculty of Medicine and Psychology, Department of Clinical and Molecular Medicine, University of Rome "Sapienza", Santo Andrea Hospital, Rome, Italy
4   Oncologic Radiotherapy Unit, Centro di Riferimento Oncologico di Aviano (CRO), IRCCS, National Cancer Institute, Aviano, Italy
5   Università Cattolica del Sacro Cuore, Fondazione Policlinico Universitario Agostino Gemelli, Polo Scienze Oncologiche ed Ematologiche, Rome, Italy
6   Division of Otorhinolaryngology, Azienda Ospedaliera Santa Maria degli Angeli, Pordenone, Italy
7   Medical Physics Unit, Centro di Riferimento Oncologico di Aviano (CRO), IRCCS, National Cancer Institute, Aviano, Italy
8   Institute of Molecular Bioimaging and Physiology, National Research Council (IBFM-CNR), Milan, Italy
9   Department of Physics, Università degli Studi di Milano-Bicocca, Milan, Italy
10  Division of Pathology, Azienda Ospedaliera Santa Maria degli Angeli, Pordenone, Italy
    *Corresponding author. Tel: +39 0434 659779; Fax: +39 0434 659429; E-mail: gbaldassarre@cro.it
    †Present address: Centro Diagnostica MINERVA, Rome, Italy

Among HNSCC patients, those with HPV-positive oro-pharyngeal cancers have a particularly good prognosis and represent a population in which de-escalation treatments would be strongly recommended. Yet, very recent trials demonstrated that RT+CTX is inferior to standard CDDP regimen in low-risk HPV-positive HNSCC patients, indicating that RT+CTX does not represent a good alternative to reduce toxicity in these patients (Gillison et al, 2019; Mehanna et al, 2019).

The identification of biomarkers able to predict patients' response to CTX is an urgent clinical need in HNSCC and could result in an immediate benefit to certain subgroups of patients, improving their survival and decreasing toxicity (Hammerman et al, 2015; Huang et al, 2016).

MicroRNA (miR) expression is altered in tumors, and their deregulation has been used for diagnosis as well as to predict relapse, survival, and response to therapies (Iorio & Croce, 2012). In HNSCC, several miRs are differentially expressed with respect to the normal/peritumoral mucosa (Babu et al, 2011; Sethi et al, 2014; Jamali et al, 2015). Recently, we have discovered that a four-miR signature (miR-1, miR-9 miR-133, and miR-150) efficiently identifies HNSCC patients at high risk of developing loco-regional recurrence (Citron et al, 2017).

These four miRs as a whole targeted the epithelial to mesenchymal transition (EMT) process, impacting on HNSCC response to therapies and, eventually, on progression (Citron et al, 2017). Cancer cell plasticity and EMT are largely regulated epigenetically (Tam & Weinberg, 2013), and miRs play a primary role in modulating these processes in HNSCC (Babu et al, 2011; Ceppi & Peter, 2014; Citron et al, 2017). Among the four miRs composing the signature, miR-9 displayed the strongest association with recurrence risk (Citron et al, 2017). Consistent with these findings, other studies recently reported that miR-9 is overexpressed in the saliva from HNSCC patients (Salazar et al, 2014), is enriched in the cancer stem cell population, and correlates with invasion and metastasis in a mouse model of SSC (White et al, 2013). However, how miR-9 regulates HNSCC aggressiveness and whether it modulates the response to therapies are unknown.

Here, we have investigated the role of miR-9 in HNSCC and provide evidences that miR-9 is involved in the response to therapies and its expression predicts poor survival in HNSCC patients treated with RT+CTX combination.

## Results

### miR-9 regulates plasticity in HNSCC-derived cells

We have previously reported that in a panel of HNSCC-derived cell lines, miR-9 is expressed at different levels, with the highest expression in FaDu and SCC9 cells (Citron et al, 2017). Here, using four TP53-mutated cell lines (Bradford et al, 2003), we confirmed these data showing that FaDu and SCC9 cell lines displayed the highest, while CAL27 and UMSCC1 cell lines displayed the lowest, miR-9 expression, comparable with the one of normal epithelial cells (NHBE) (Appendix Fig S1A and B). These cell lines have then been used to characterize the role of miR-9 in HNSCC cell growth and response to therapies.

*In vivo*, FaDu cells (miR-9 high) grew much faster than CAL27 cells (miR-9 low), even when a lower number of cells was injected

$(1 \times 10^6$ FaDu vs. $5 \times 10^6$ CAL27) (Appendix Fig S1C). Based on these data and on new evidences indicating that miR-9 regulates cell plasticity and cancer stem cell-like phenotypes in a model of SCC (White et al, 2013), we thus investigated the role of miR-9 in tumor-initiating properties of HNSCC cells.

To this aim, we generated stable FaDu cells expressing the anti-miR-9 or an empty vector (shCTR) (Fig 1A). miR-9 reduction resulted in increased expression of SASH1 and KRT13, already identified as miR-9 targets in HNSCC cells (Citron et al, 2017), and of the epithelial marker ZO-1 (Fig 1B). These changes were accompanied by a shift in cell morphology toward a more epithelial shape, both under basal conditions and after stimulation with TGF-β (Appendix Fig S1D). Cell motility was also slightly decreased in anti-miR-9 compared with control FaDu cells (Appendix Fig S1E and F). These differences were accompanied by a strong decrease in the ability to grow both in anchorage-dependent manner and in anchorage-independent manner, as demonstrated by the growth curve, clonogenic, and soft agar assays, respectively (Fig 1C–E). More importantly, anti-miR-9 FaDu cells displayed lower sphere-forming and self-renewal abilities, in terms of both number and size of the spheres (Fig 1F). Consistent with these in vitro data, upon transplantation in animals, anti-miR-9 FaDu cells formed smaller tumors with longer latency, compared to shCTR cells (Fig 1G and H). Further, tumors grown from anti-miR-9 FaDu cells showed a lower percentage of Ki67-positive cells, compared to those from shCTR cells (Fig 1I). On the contrary, when we stably overexpressed miR-9, CAL27 cells lost the expression of SASH1, KRT13, and ZO-1, and acquired increased proliferation potential, accompanied by higher colony- and sphere-formation capabilities (Fig EV1A–E). Accordingly, upon transplantation in animals, miR-9 overexpressing CAL27 cells formed bigger tumors and shorter latency compared with controls, when more challenging conditions were used (i.e., injection of $0.5 \times 10^6$ and $1 \times 10^6$ cells) (Fig EV1F and G). Histological analyses of the miR-9 overexpressing tumors revealed a higher number of Ki67-positive cells and a less differentiated phenotype, characterized by the absence of corneal pearls, compared to controls (Fig. EV1H). These findings were confirmed in SCC9 cells (high miR-9), by downmodulation of miR-9, and UMSCC1 cells (low miR-9), by overexpression of miR-9, characterized for their in vitro behavior (Appendix Fig S2A–I). Altogether, results showed that low miR-9 level resulted in higher expression of SASH1, KRT13, and ZO-1 and lower proliferation and survival potential (Appendix Fig S2A–I). Strikingly, anti-miR-9 CAL27 cells, that displayed almost undetectable levels of miR-9, showed a slight increase in SASH1, KRT13, and ZO-1 (Appendix Fig S2J and K) and completely failed to engraft and grow in immunodeficient mice (Appendix Fig S2L).

### miR-9 expression is positively regulated by EGFR activation in HNSCC cells

Results collected so far clearly indicated that miR-9 expression strongly regulated HNSCC aggressive phenotype. We next asked how miR-9 was regulated in HNSCC cells. Literature data indicated that miR-9 expression correlates with cell proliferation and is positively regulated by the activation of RAS and c-Myc in breast cancer (Ma et al, 2010), suggesting that mitogenic stimuli may positively regulate miR-9 transcription also in HNSCC. In accord with this hypothesis, serum stimulation increased by threefold the promoter

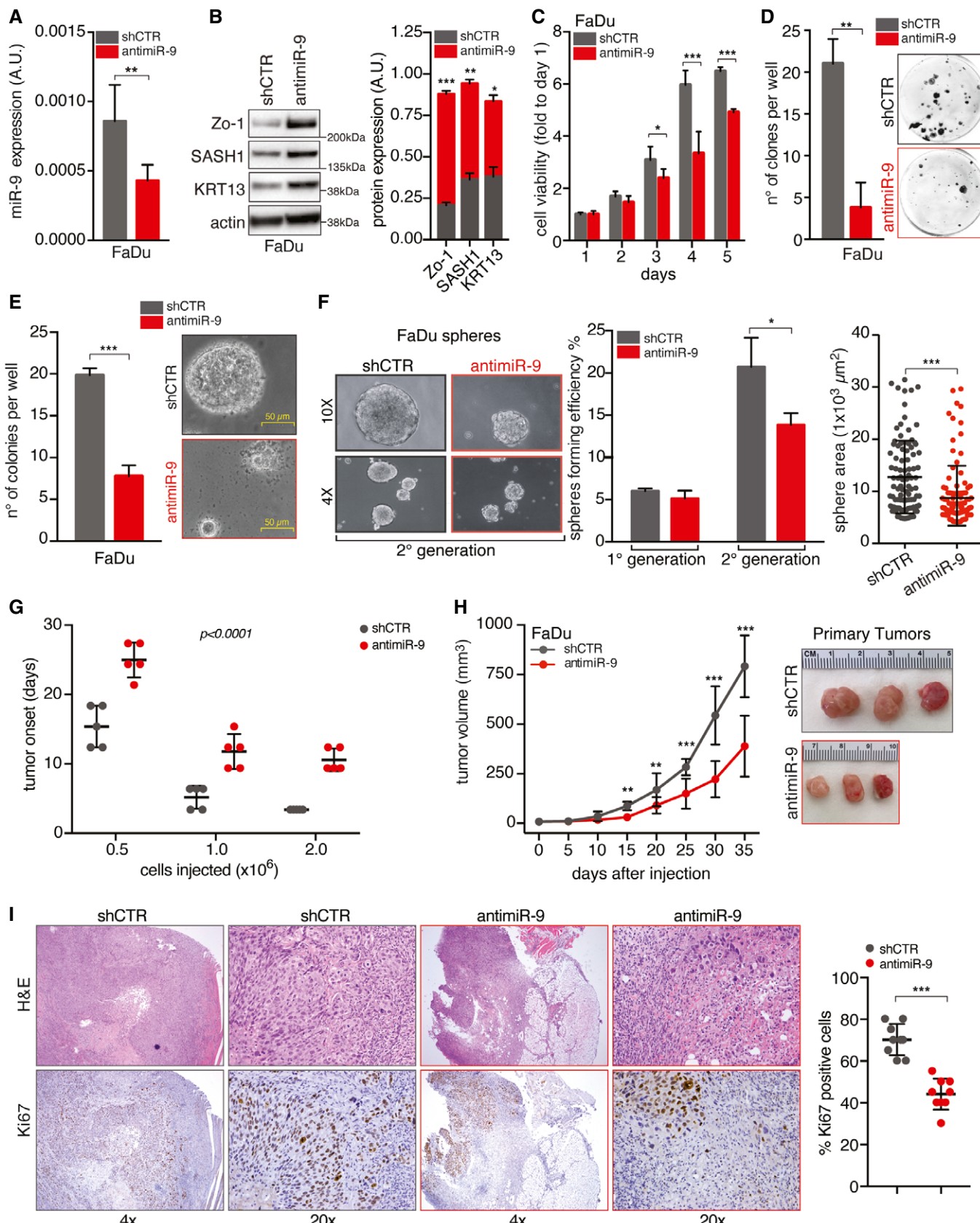

**Figure 1.**

**Figure 1.  miR-9 expression sustains the tumor-initiating properties of HNSCC cells.**

A  qRT–PCR analyses of miR-9 expression in control (shCTR) and anti-miR-9 FaDu cells. Data represent the mean (± SD) of three independent experiments performed in duplicate, and unpaired *t*-test was used to verify the statistical significance.

B  On the right, Western blot (WB) analyses of the indicated protein expression in FaDu cells as in (A). Actin was used as loading control. Right graph reports the quantification of protein expression normalized on actin loading control. Data represent the mean (± SD) of three independent experiments. Unpaired *t*-test was used to verify the statistical significance.

C  Cell viability analyses of cells described in (A, B) over a period of 5 days using the MTS assay. Data represent the mean (± SD) of three independent experiments performed in sextuplicate. Unpaired *t*-test was used to verify the statistical significance per each time point.

D  Colony formation assay of the cells described in (A, B). Left graph reports the number of clones per well. Data represent the mean (± SD) of three independent experiments performed in duplicate, and unpaired *t*-test was used to verify the statistical significance. Right, representative images of clones are shown.

E  Soft agar assay of cells described in (A, B). Left graph reports the number of clones per well. Data represent the mean (± SD) of three independent experiments performed in duplicate, and unpaired *t*-test was used to verify the statistical significance. Right, representative images of 10× field are shown.

F  Sphere-forming assay of cells described in (A, B). Left, typical images of 4× and 10× fields are shown. Middle graph reports the number of spheres formed in first and second generations. Right graph reports the area of second-generation spheres. Each dot represents one analyzed sphere. Data represent the mean (± SD) of three independent experiments performed in duplicate, and unpaired *t*-test was used to verify the statistical significance.

G  Graph reporting the tumor onset in NSG mice injected with different numbers of control (shCTR) and anti-miR-9 FaDu cells, as described, and followed for up to 35 days (*n* = 5 mice/group). Data represent the mean (± SD), and two-way ANOVA test was used to calculate the statistical significance among groups.

H  Left graph reports the tumor volume in NSG mice injected with control (shCTR) and anti-miR-9 FaDu cells followed for up to 35 days (*n* = 5 mice/group). Data represent the mean (± SD), and unpaired *t*-test was used to verify the statistical significance at each time point. On the right, typical images of explanted tumors formed by control (shCTR) and anti-miR-9 FaDu cells at necropsy.

I  Typical images of hematoxylin & eosin (H&E—upper images) and Ki67 expression (Ki67—bottom images) evaluated by immunohistochemistry (IHC) in tumors explanted from mice treated as in (H). 4× and 20× magnification are shown. Right graph reports the percentage of Ki67-positive cells. Each dot represents one tumor. Data are expressed as mean (± SD) and unpaired *t*-test was used to verify the statistical significance.

Data information: In the figure, A.U. = arbitrary units, *P < 0.05; **P < 0.01; ***P < 0.001.
Source data are available online for this figure.

activity of miR-9, starting at 1 h from serum addition (Fig 2A). EGFR is the most frequently amplified/mutated growth factor receptor in primary HNSCC (The Cancer Genome Atlas Network, 2015), whose biological role has represented the rationale for the design of targeted anti-EGFR treatments in combination with RT (Santuray *et al*, 2018). We thus tested whether EGFR activation regulated miR-9 expression. EGF stimulation increased by threefold the expression of endogenous miR-9 (Fig 2B), while inhibition of EGFR with gefitinib, a small-molecule inhibitor, or Cetuximab (CTX), a monoclonal antibody approved for the treatment of HNSCC patients in combination with RT (Santuray *et al*, 2018), decreased the activity of miR-9 promoter and reduced the expression of endogenous miR-9, in both control and anti-miR-9 FaDu cells (Fig 2C–F).

Given the relevance of EGFR amplification/mutation in HNSCC, we asked whether miR-9 could interfere with the response to anti-EGFR-targeted therapy. Indeed, miR-9 silencing sensitized FaDu and SCC9 cells to EGFR inhibition, *in vitro* (Fig 2G and H, Appendix Fig S3A), while miR-9 overexpression protected UMSCC1 and CAL27 from CTX-induced cell death (Appendix Fig S3B and C), supporting that miR-9 could be implicated in response to anti-EGFR treatments. Furthermore, EGFR and miR-9 expression levels significantly correlated in HNSCC samples (Fig 2I and Appendix Table S1).

**miR-9 positively regulates Sp1 expression *via* KLF5 downmodulation in HNSCC cells**

To explain how miR-9 may possibly regulate the tumorigenic potential of HNSCC cells, we first looked to the oncosuppressor SASH1 that we previously showed to be a *bona fide* miR-9 target in HNSCC (Citron *et al*, 2017) and has been involved in the regulation of EMT in different tumor types, including cutaneous SCC (He *et al*, 2016; Chen *et al*, 2020; Franke *et al*, 2020). As expected, modulation of miR-9 resulted in altered SASH1 protein expression (Figs 1B and EV1B and Appendix Fig S2B,G,K). However, modification of SASH1

expression did not alter the *in vitro* properties of either FaDu or CAL27 cells, inducing no significant change in proliferation, colony formation, motility, or sphere formation (Appendix Fig S4A–F). Similarly, silencing of SASH1 expression in CAL27 cells only slightly increased their proliferation, but was not accompanied by any change in colony formation or migration abilities (Appendix Fig S4G–J). Overall, these data suggested that miR-9 regulated cell proliferation and survival of HNSCC cells in a SASH1-independent manner.

In our previous study, we observed that miR-9 positively correlated with the expression of Sp1 in HNSCC patients included in the TCGA dataset (Citron *et al*, 2017). Sp1 is a transcription factor that has been linked to both the acquisition of stem-like properties and the resistance to radiotherapy, through the alteration of the DNA damage response (DDR) in different types of cancer, including HNSCC (Olofsson *et al*, 2007; Beishline *et al*, 2012; Tschaharganeh *et al*, 2014; Zhang *et al*, 2014; Beishline & Azizkhan-Clifford, 2015; Liu *et al*, 2016; Xu *et al*, 2017; Fletcher *et al*, 2018).

First, we validated the observation made in the TCGA dataset in a consecutive series of 150 primary HNSCC samples, collected by our surgeons (Fig 3A and Appendix Table S1). Then, we investigated whether miR-9 regulated the expression of Sp1 in our *in vitro* models. In FaDu cells, both anti-miR-9 and miR-9-sponge expression decreased the levels of Sp1 mRNA (Figs 3B and EV2A), while overexpression of miR-9 increased Sp1 mRNA levels in control FaDu cells, but not in the ones stably expressing anti-miR-9 (Fig 3C). These results were confirmed in CAL27 cells, in which miR-9 overexpression, via either lentiviral or retroviral transduction (i.e., miR-9 or GFP-miR-9), resulted in a strong upregulation of Sp1 mRNA (Figs 3D and EV2B).

To verify whether Sp1 downmodulation could recapitulate the phenotype observed in anti-miR-9 cells, we generated Sp1-silenced FaDu and CAL27 cells, using a lentiviral shRNA approach. In these settings, Sp1 silencing strongly impaired the proliferation,

clonogenic, and spherogenic abilities of FaDu and CAL27 cells (Fig 3 E and F, and Appendix Fig S5A–D). The results were confirmed using low doses of mithramycin A, a drug that is able to globally displace Sp1 from DNA by binding GC-rich DNA domains (Appendix Fig S5E and F).

To evaluate whether Sp1 could be involved in the response to anti-EGFR therapies, we overexpressed Sp1 in FaDu cells silenced or not for miR-9 expression (anti-miR-9) (Fig 3G). Strikingly, Sp1 expression was sufficient to completely rescue the sensitivity to

EGFR blockade in FaDu anti-miR-9 cells, but not in FaDu shCTR cells (Fig 3H). These data were also confirmed in shCTR and anti-miR-9 CAL27 cells, overexpressing or not Sp1 (Fig EV2C and D), supporting that Sp1 may represent a major contributor to the miR-9-mediated resistance to EGFR blockade observed in HNSCC.

To understand how miR-9 could regulate Sp1 expression, we cloned the Sp1 promoter (Nicolás *et al*, 2001) in a luciferase reporter vector (Fig 3I). We had observed that miR-9 increased Sp1 mRNA (Fig 3B–D) and, accordingly, stable miR-9 overexpression in CAL27

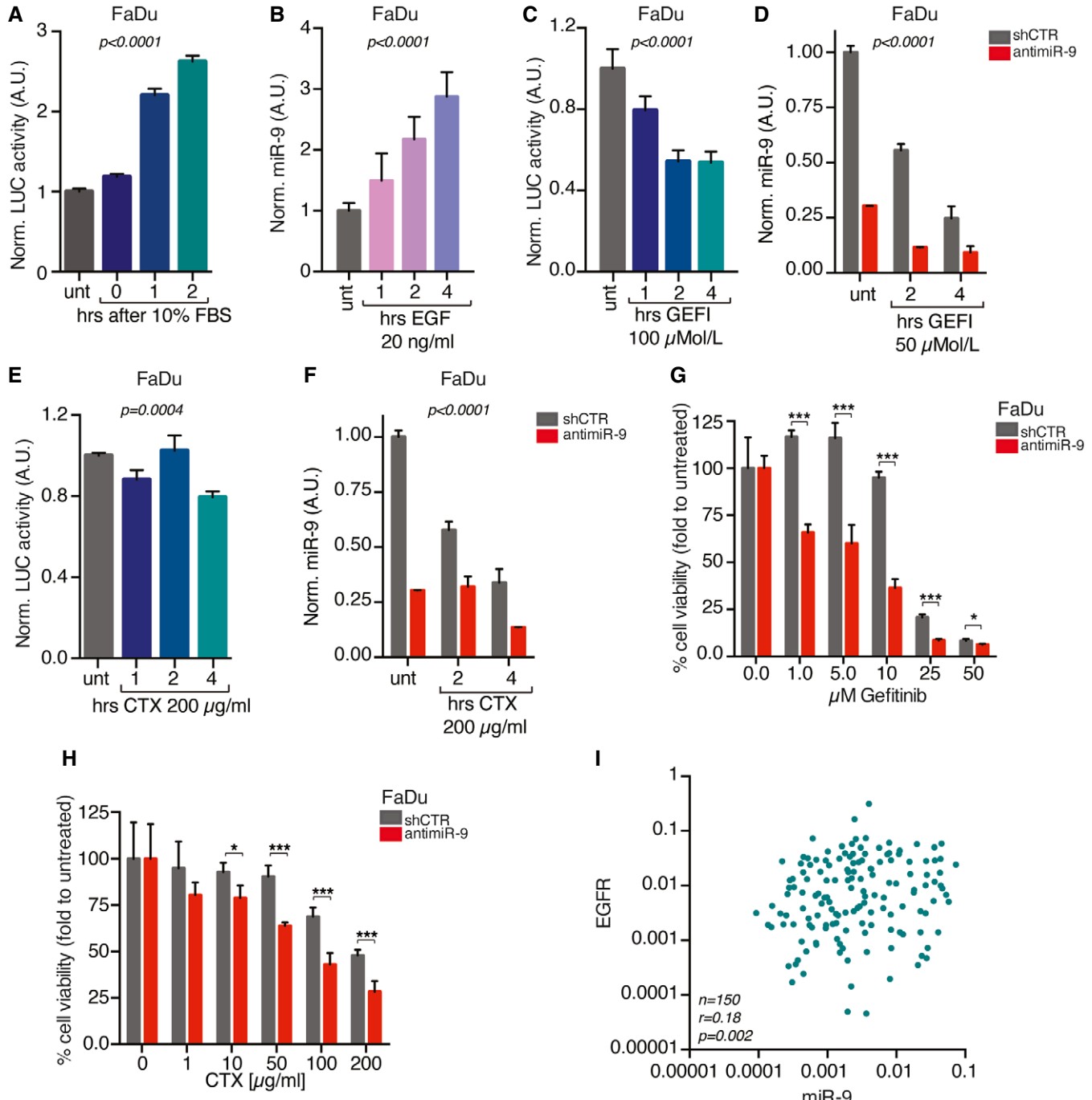

Figure 2.

◀

**Figure 2. miR-9 expression is induced by EGFR pathway activation in HNSCC.**

A  Graph reporting the normalized luciferase activity of miR-9 promoter, expressed as fold over the untreated condition, in FaDu cells serum starved and then stimulated with 10% fetal bovine serum (FBS) for up to 2 h. Data represent the mean (± SD) of three independent experiments performed in duplicate, and one-way ANOVA test was used to verify the statistical significance.

B  qRT–PCR analyses of normalized miR-9 expression, expressed as fold over the untreated condition, in FaDu cell serum starved and then stimulated with 20 ng/ml of epidermal growth factor (EGF) for up to 4 h. Data represent the mean (± SD) of three independent experiments performed in duplicate, and one-way ANOVA test was used to verify the statistical significance.

C  Graph reporting the normalized luciferase activity of miR-9 promoter, expressed as fold over the untreated condition, in FaDu cells treated with gefitinib (GEFI) for up to 4 h. Data represent the mean (± SD) of three independent experiments performed in duplicate, and one-way ANOVA test was used to verify the statistical significance.

D  qRT–PCR analyses of normalized miR-9 expression, expressed as fold over the untreated condition, in control (shCTR) and anti-miR-9 FaDu cells treated with gefitinib (GEFI) for up to 4 h. Data represent the mean (± SD) of three independent experiments performed in duplicate, and two-way ANOVA test was used to verify the statistical significance.

E  Graph reporting the normalized luciferase activity of miR-9 promoter, expressed as fold over the untreated condition, in FaDu cells treated with Cetuximab (CTX) for up to 4 h. Data represent the mean (± SD) of three independent experiments performed in duplicate, and one-way ANOVA test was used to verify the statistical significance.

F  qRT–PCR analyses of normalized miR-9 expression, expressed as fold over the untreated condition, in control (shCTR) and anti-miR-9 FaDu cells treated with Cetuximab (CTX) for up to 4 h. Data represent the mean (± SD) of three independent experiments performed in duplicate, and two-way ANOVA test was used to verify the statistical significance.

G, H  Graph reporting the cell viability of control (shCTR) and anti-miR-9 FaDu cells treated with increasing concentration of gefitinib (G) or Cetuximab (CTX) (H) as indicated and evaluated using the MTS assay. Data represent the mean (± SD) of two independent experiments performed in sextuplicate, and unpaired *t*-test was used to verify the statistical significance per each dose.

I  Dot plot reporting the correlation of EGFR and miR-9 expression in primary HNSCC samples evaluated by qRT–PCR. The number of analyzed samples (*n*), the Spearman correlation value (*r*), and its significance (*P*) are reported in the graph.

Data information: In the figure, A.U. = arbitrary units *$P < 0.05$; ***$P < 0.001$.

cells markedly increased the promoter activity of Sp1 respect to the control, supporting that miR-9 positively regulated the transcription of Sp1 (Fig EV2E). These data were confirmed in FaDu cells, in which the abrogation of miR-9 (anti-miR-9) strongly reduced the Sp1 promoter activity (Fig 3J). However, neither Sp1 promoter sequence nor its 3' UTR region contained any miR-9 seed sites, suggesting that miR-9 affected SP1 transcription indirectly, through the regulation of a different target gene. Using a bioinformatic approach (Pavón *et al*, 2012; Citron *et al*, 2017), we identified 20 genes possibly targeted by miR-9 and downregulated during HNSCC progression (Fig EV3A). We analyzed the expression of these genes in FaDu control and anti-miR-9 cells by qRT–PCR and found that two of them were not detectable, eleven not modified, and seven were upregulated in anti-miR-9 FaDu cells (Fig 4A and B). Among these, we focused our attention on KLF5 because several putative KLF5-binding sites are present in the Sp1 promoter region (Fig 3I) and because a KLF5 deletion was already linked to Sp1 upregulation in a model of prostate cancer progression (Xing *et al*, 2014). Further, KLF5 often acts as tumor suppressor and its loss has been involved in several aspects of cancer progression, including tumor initiation (Tetreault *et al*, 2013; Farrugia *et al*, 2016). We corroborated our data using SCC9, UMSCC1, and CAL27 cells and observed that inhibition of miR-9 in SCC9 cells strongly upregulated KLF5 mRNA and protein expression and that overexpression of miR-9 in UMSCC1 and CAL27 cells reduced KLF5 mRNA and protein levels (Fig EV3B–D).

Next, we tested whether miR-9 could directly regulate KLF5 expression acting on its 3'-UTR, which contains two different seed sites for miR-9 (Fig 4C). Luciferase assay in FaDu and CAL27 cells demonstrated that miR-9 knockdown significantly increased and miR-9 overexpression reduced the luciferase activity, when both the seed sites were present (WT) (Figs 4D and EV3E). On the contrary, when both the seed sites in 3'UTR of KLF5 were mutated (mut A + B), miR-9 modification failed to modulate KLF5-driven LUC activity in CAL27 cells (Fig EV3E) and reduced the luciferase

activity in FaDu cells, compared to the single mutants (Fig 4D). Overall, these data demonstrated that KLF5 represents a *bona fide* target of miR-9 in HNSCC cells, as recently reported in HEK293 cells for the rat KLF5 gene (Yang *et al*, 2019).

We next tested whether KLF5 could regulate Sp1 expression in HNSCC. Overexpression of KLF5 in FaDu cells resulted in a strong down-regulation of Sp1 mRNA and protein levels (Fig 4E and F). This result was corroborated by the reduced Sp1 promoter activity after overexpression of KLF5, in both FaDu and CAL27 cells (Figs 4 G and EV3F). Chromatin immunoprecipitation (ChIP) assay on FaDu and CAL27 cells confirmed that endogenous KLF5 bound to the Sp1 promoter on four possible binding sites, located between base –253 and –1602 from the ATG (Figs 4H and EV3G). Positive and negative controls demonstrated the specificity of this binding (Fig EV3H).

Altogether, data collected so far strongly indicate that miR-9 regulates Sp1 by targeting KLF5 and participates to the tumorigenic potential of HNSCC cells and to the resistance to EGFR blockade.

**miR-9 regulates the response to radiotherapy but not to chemotherapy in HNSCC cells**

Next, we tested whether miR-9 expression was implicated in tumor response to standard of care regimens (chemotherapy and radiotherapy) used in the management of advanced HNSCC patients. To address this point, we used FaDu and SCC9 cells (high miR-9) transduced with anti-miR-9 or control vector and CAL27 cells (low miR-9) to overexpress miR-9 or control vector. To mimic relevant clinical settings, we tested cisplatin (CDDP), 5-fluorouracil (5-FU), paclitaxel (TAX), and the radiomimetic drug bleomycin. When administered as single agents, only bleomycin showed a different efficacy in dependence on miR-9 expression, being more effective in FaDu and SCC9 anti-miR-9 cells and less in miR-9 overexpressing CAL27 cells, compared to respective control (Figs 5A and EV4A, Appendix Fig

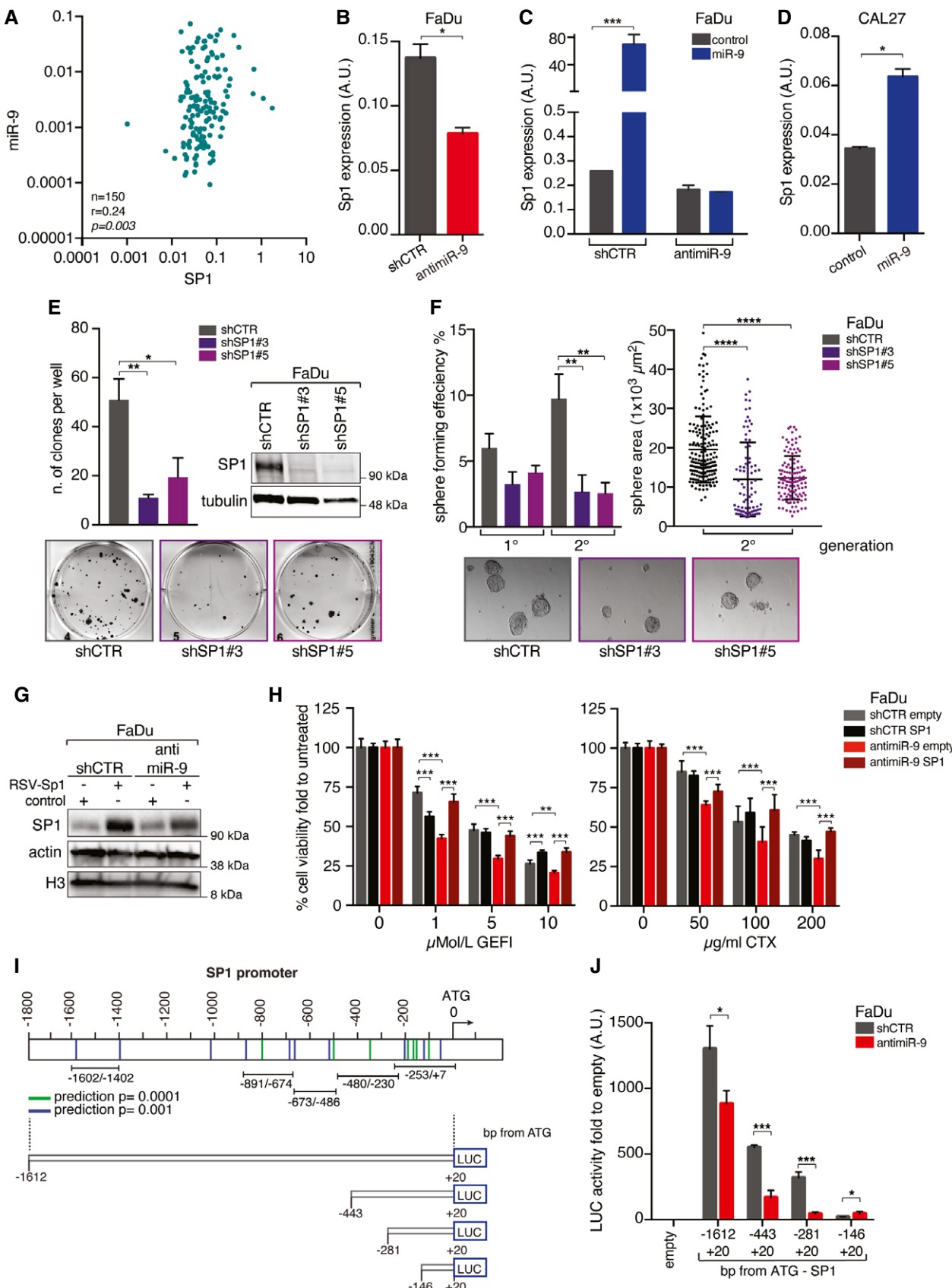

**Figure 3.**

**Figure 3. Sp1 expression is positively regulated by miR-9.**

A   Dot plot reporting the correlation of miR-9 and Sp1 expression in primary HNSCC samples evaluated by qRT–PCR. The number of analyzed samples (*n*), the Spearman correlation value (*r*), and its significance (*P*) are reported in the graph.

B   qRT–PCR analyses of SP1 expression in control (shCTR) and anti-miR-9 FaDu cells. Data represent the mean (± SD) of three independent experiments performed in duplicate, and unpaired *t*-test was used to verify the statistical significance.

C   qRT–PCR analyses of Sp1 expression in control (shCTR) and anti-miR-9 FaDu cells transiently transduced with PGK-miR-9 or control vector as indicated. Data represent the mean (± SD) of three independent experiments performed in duplicate, and two-way ANOVA with Sidak's multiple comparison test was used to verify the statistical significance.

D   qRT–PCR analyses of Sp1 expression in CAL27 cells transfected with pcDNA miR-9 or control vector. Data represent the mean (± SD) of three independent experiments performed in duplicate, and unpaired *t*-test was used to verify the statistical significance.

E   Clonogenic assay on FaDu cells silenced for Sp1 (shSP1#3 or #5) or scramble sequence (shCTR). Left graph reports the number of colonies. On the right, WB analyses of the indicated protein expression in FaDu cells. Tubulin was used as loading control. Bottom panel, representative images of clones are shown. In the graph, data represent the mean (± SD) of three independent experiments performed in duplicate and one-way ANOVA was used to verify the statistical significance.

F   Sphere-forming assay of the cells described in (E). Left panel reports the mean number of spheres formed in first and second generations. Right graph reports the area of second-generation spheres. Data represent the mean (± SD) of three independent experiments performed in duplicate, and one-way ANOVA was used to verify the statistical significance. Each dot represents one analyzed sphere. Bottom panels show representative images of the spheres.

G   WB analyses of the indicated protein expression in control (shCTR) and anti-miR-9 FaDu cells, overexpressing or not RSV-SP1. Actin and histone H3 were used as loading control.

H   Graphs reporting the cell viability of FaDu cells described in G and treated with increasing concentration of gefitinib (GEFI—left panel) or Cetuximab (CTX—right panel) as indicated and evaluated using the MTS assay. Data represent the mean (± SD) of two independent experiments performed in sextuplicate, and two-way ANOVA with Sidak's multiple comparison test was used to verify the statistical significance.

I   Schematic representation of the pGL3 vectors used to test the potential activity miR-9 on Sp1 promoter. Vectors are named based on the maximum distance from SP1 ATG. The lines indicate the putative KLF5-binding sites according to the statistical prediction power according to EPD portal (blue lines *P* = 0.001, green lines *P* = 0.0001). Black bars indicate the primer pairs used for the amplification of different region of Sp1 promoter in the ChIP analyses depicted in Fig 4H.

J   Graph reporting the normalized luciferase activity of Sp1 promoter fragments in control (shCTR) or anti-miR-9 FaDu cells. Data represent the mean (± SD) of three independent experiments performed in duplicate, and unpaired *t*-test was used to verify the statistical significance.

Data information: In the figure, A.U. = arbitrary units *$P < 0.05$; **$P < 0.01$; ***$P < 0.001$; ****$P < 0.0001$.
Source data are available online for this figure.

S6A). This result suggested a specific role for miR-9 in protecting from radiation-induced cell death. To validate this finding, we irradiated (IR) cells and confirmed that miR-9 expression protected them from IR-induced death, in both FaDu and SCC9 cells (Figs 5B and EV4B). In accord with these data, CAL27 cells stably overexpressing miR-9 were more resistant to IR-induced death than controls (Appendix Fig S6B).

To verify whether miR-9 could impact on the IR-induced DNA damage response (DDR), we assessed the expression of γH2AX (DNA damage marker) in FaDu and SCC9 (miR-9 high) miR-9 silenced cells and in CAL27 and UMSCC1 (miR-9 low) miR-9 overexpressing cells, compared to respective control. We irradiated cells with 2 or 5 Gy, depending on the sensitivity of each cell line, and allowed them to repair the DNA damage for 8–24 h. A rapid and more sustained expression of γH2AX was observed when miR-9 expression was lower than controls, in all tested cell lines (Figs 5C and EV4C, Appendix Fig S6C). These data were also confirmed by immunofluorescence analyses of γH2AX and pSer10-H3 (marker of mitosis), in FaDu and CAL27 cells modified for miR-9 expression, treated as above. miR-9 silencing did not affect the M phase, but increased the number of damaged cells (γH2AX positive) and the time necessary to repair the damage, supporting a role for miR-9 in the response to DNA damage following IR (Fig 5D). Accordingly, overexpression of miR-9 significantly reduced the number of γH2AX-positive cells and improved the recovery after the IR-induced cell cycle arrest, measured as % of pSer10-H3-positive cells (Appendix Fig S6D), corroborating a possible impact of miR-9 on DDR following IR. Interestingly, Sp1 expression paralleled miR-9 levels in irradiated FaDu cells, showing a reduction in both protein and mRNA levels (Fig 5E). Moreover, Sp1 protein was downregulated in both control and anti-miR-9 FaDu cells and its mRNA expression did not recover up to 24 h after IR, an effect more

evident in control cells that expressed higher basal levels of Sp1 mRNA (Figs 5C and E, and EV4C).

Next, we investigated whether Sp1 overexpression could be involved in resistance to IR. Using the radiomimetic agent bleomycin, we observed that forced Sp1 expression strongly increased resistance to bleomycin in control cells and reverted the IR sensitivity of anti-miR-9 FaDu and CAL27 cells (Fig 5F, Appendix Fig S6E).

These data further support that miR-9/Sp1 axis is critical to sustain radio-resistance in HNSCC cells. However, the same was not true for CDDP treatment, suggesting that miR-9 could act in the response to double- or single-strand break induced by IR but not to inter- and intra-strand cross-linking induced by CDDP (Weber & Ryan, 2015; Gavande *et al*, 2016). To test this hypothesis, we either irradiated or CDDP-treated CAL27 cells (control and miR-9) and looked at γH2AX expression. In accord with our hypothesis, γH2AX levels were reduced in miR-9 expressing cells treated with IR but not in those treated with CDDP (Appendix Fig S6C and H). Moreover, we observed that miR-9 levels transiently decreased after IR, while they slightly increased after CDDP treatment (Fig 5E, Appendix Fig S6G). To understand whether this modulation of miR-9 was due to a transcriptional control, we used a luciferase reporter vector under the control of miR-9 promoter (Ma *et al*, 2010) and observed that the radiomimetic drug bleomycin significantly reduced miR-9 promoter activity, in a time-dependent manner (Appendix Fig S6H).

**miR-9 regulates the response to IR+CTX and is a potential biomarker for the response to this combination treatment in HNSCC patients**

The data collected so far demonstrated that high levels of miR-9 correlated with resistance to both anti-EGFR and IR treatments, used as single agents in HNSCC cells. Toward a clinical translation of our

results, we next verified whether these data were also maintained in *in vivo* settings. We subcutaneously injected mice with control and anti-miR-9 FaDu cells. After tumor appearance, mice were randomized into two groups: vehicle and CTX (1 mg/kg), administered two times a week, for three weeks. A strong effect of miR-9 was observed, and tumor appearance was significantly delayed in

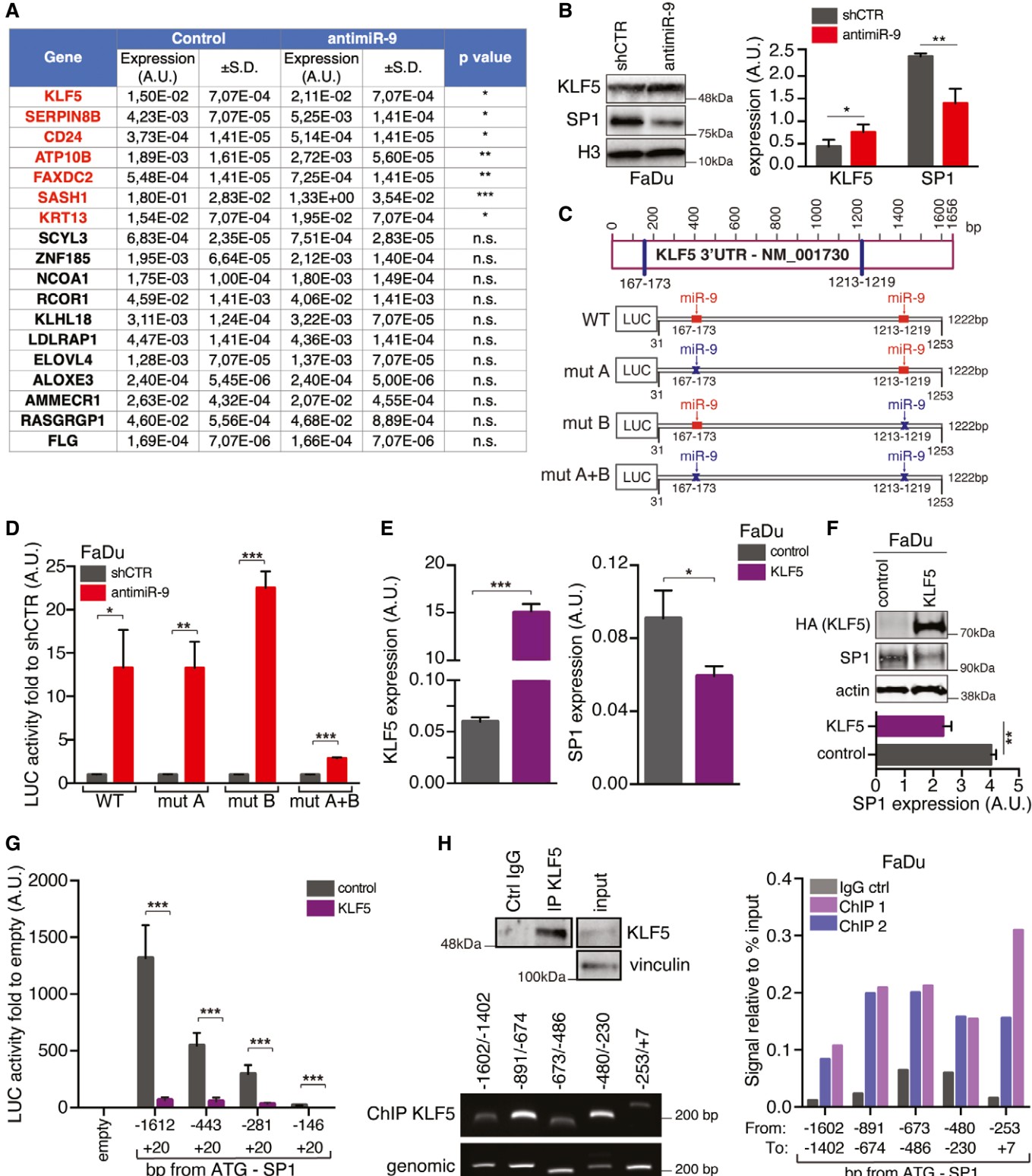

Figure 4.

**Figure 4.  miR-9 regulates Sp1 expression by targeting KLF5 transcription factor.**

A   Table reporting the qRT–PCR analyses of gene expression in control (shCTR) and anti-miR-9 FaDu cells, as indicated. Data represent the mean ($\pm$ SD) of three independent experiments performed in duplicate, and unpaired *t*-test was used to verify the statistical significance.

B   On the left, WB analyses of KLF5 and Sp1 expression in control (shCTR) and anti-miR-9 FaDu. Histone H3 expression was used as loading control. On the right, protein quantification analyses of KLF5 and SP1 expression normalized on H3 loading control. Data represent the mean ($\pm$ SD) of three biological replicates, and unpaired *t*-test was used to verify the statistical significance.

C   Schematic design of the KLF5 3'-UTR. To test the potential miR-9 binding on KLF5 3'-UTR, four vectors were generated: WT (wild type containing both the seed sites), mut A or mut B (mutated for one single binding site) and mut A + B (mutated in both seed site). The predicted seed sites for miR-9-binding sites are shown in red when present and in blue when mutated.

D   Graph reporting the normalized luciferase activity of wild type (WT) or mutated KLF5 3'-UTR described in C in control (shCTR) or anti-miR-9 FaDu cells, as indicated. Data represent the mean ($\pm$ SD) of three independent experiments performed in duplicate, and unpaired *t*-test was used to verify the statistical significance.

E   qRT–PCR analyses of KLF5 (left) and Sp1 (right) expression in FaDu cells transfected with control or KLF5 vectors. Data represent the mean ($\pm$ SD) of three independent experiments performed in duplicate, and unpaired *t*-test was used to verify the statistical significance.

F   Upper panel shows WB analyses of KLF5 and Sp1 expression in FaDu cells. Actin expression was used as loading control. Lower graph shows the SP1 protein quantification normalized over actin expression. Data represent the mean ($\pm$ SD) of three independent experiments, and unpaired *t*-test was used to verify the statistical significance.

G   Graph reporting the normalized luciferase activity of Sp1 promoter fragments in FaDu transfected with pcDNA control or KLF5 vectors, as indicated. Data represent the mean ($\pm$ SD) of three independent experiments performed in duplicate, and unpaired *t*-test was used to verify the statistical significance.

H   Chromatin immunoprecipitation (ChIP) assay performed on FaDu cells. Upper left panel shows WB analysis reporting KLF5 expression in the immunoprecipitation achieved using anti-KLF5 or control (IgG) antibodies used in the ChIP assay. Vinculin expression was used as loading control. Bottom left panel shows a typical image of amplified PCR fragments using ChIP DNA or genomic DNA, as indicated. Right graph reports the binding of KLF5 to the indicated region of Sp1 promoter expressed as signal relative to input in two independent immunoprecipitations (ChIP1 and ChIP2) using the KLF5 ab. IgG was used as negative control on the same chromatin. Primer pairs used to amplify the different regions of Sp1 promoter are depicted in the schema provided in Fig 3I.

Data information: In the figure, A.U. = arbitrary units; *P < 0.05; **P < 0.01; ***P < 0.001.
Source data are available online for this figure.

anti-miR-9 FaDu cells. As already observed in clinics with HNSCC patients, CTX alone was not able to reduce the *in vivo* growth of FaDu cells, either control or anti-miR-9 (Fig EV5A), in contrast with what observed *in vitro* (Fig 2G and H, Appendix Fig S3). Nevertheless, Western blot analyses confirmed a mild activity of CTX in reducing EGFR and ERK phosphorylation compared with vehicle (Fig EV5B and C). Interestingly, immunostaining of primary tumors formed by anti-miR-9 FaDu cells showed a reduced Sp1 nuclear intensity compared with controls (Fig EV5D), confirming *in vivo* what observed *in vitro* at molecular level (Fig 3).

To further analyze the effects of miR-9 on tumor growth and response to therapies, we generated high-titer lentiviral particles encoding for anti-miR-9 or control and injected them intra-tumorally when masses formed by FaDu parental cells reached ~ 50 mm$^3$ of volume. Mice were then subjected to four cycles of IR+CTX or IR alone, as depicted in Fig 6A. In line with the pivotal role observed for miR-9 in the regulation of tumor latency and tumor-initiating properties (Figs 1 and EV1), we did not observe significant differences in the growth of these already established tumors when anti-miR-9 lentiviral particles were injected compared with the control counterpart (Fig 6B, compare black and gray curves). However, when tumors were then treated with IR and IR+CTX, injection of anti-miR-9 lentiviral particles significantly improved the efficacy of both treatments (Fig 6B). At necroscopy, measurement of miR-9 expression in explanted tumors showed that miR-9 levels were effectively reduced by intra-tumoral injection of anti-miR-9 lentiviral particles (Fig EV5E). Necrosis (by H&E) and apoptosis (by TUNEL assay) were increased and proliferation (by Ki67 IHC) decreased in anti-miR9-treated tumors, both after IR alone and IR+CTX, but especially in the IR+CTX group (Figs 6C–F, and EV5E and F). Overall, these findings demonstrate that the activation of EGFR/miR-9/Sp1 axis may represent a strong limitation for the efficacy of IR+CTX.

We next aimed to verify whether these data were also relevant to the human pathology. We interrogated the TCGA dataset and identified 31 HNSCC patients treated with RT+CTX combination therapy, for which clinical data were available. Using the upper quartile (i.e., > 75,819 reads) as cut-off, high miR-9 expression was significantly associated with a poor prognosis ($P = 0.00123$) also in the TCGA cohort (Fig 6G). Intriguingly, the same interrogation carried out in HNSCC patients treated with RT+platinum compounds (TCGA, $n = 133$, including cisplatin, carboplatin, and oxaliplatin) indicated that high miR-9 levels predicted better prognosis in this group of patients (Fig EV5G, $P = 0.002$).

We then corroborated these *in silico* data with analyses of samples collected in our Institutions. Using a droplet digital PCR (ddPCR) approach, we evaluated miR-9 expression in tumor biopsies retrospectively collected from a cohort of patients who were treated with RT+CTX therapy in our Institute between 2010 and 2016 ($n = 16$) and from a second cohort of patients treated at the University Cattolica/Gemelli of Rome and followed up for at least 7 years ($n = 21$) (Appendix Table S2). Two samples were excluded due to low RNA quality, and remaining ones ($n = 35$) were clustered according to miR-9 expression, above (high) or below (low) the median expression of miR-9 across the entire cohort. Although the population of patients analyzed was relatively small, the expression of miR-9 represented a very strong predictor of prognosis in this setting (HR 3.75–0.27, $P = 0.0382$) (Fig 6H).

Overall, these data support the possibility that evaluation of miR-9 expression in primary HNSCC tumors could be used as biomarker to guide physicians choosing between the most appropriate treatment, CTX or platinum, to be combined with RT in HNSCC patients.

## Discussion

Here, we describe a new signaling axis, involving EGFR, miR-9, KLF5, and Sp1, that connects the tumor stem-like features of HNSCC with the response to therapy. The central node of this pathway is

miR-9 that can indirectly control Sp1 transcriptional activity through KLF5 and thereby regulates the response to DNA damage and the biological behavior of HNSCC cells.

We observed that miR-9 negatively regulates the expression of KLF5, targeting two seed sites in its 3'-UTR, both conserved along the evolution. Attenuated levels of KLF5 then release Sp1 promoter repression and enhance Sp1 transcription and expression,

unleashing its tumorigenic transcriptional activities. It has been proposed that KLF5 can act either as tumor suppressor or oncogene, depending on the tissue and cellular context (Tetreault *et al*, 2013). In a TP53-mutated genetic context, KLF5 predominantly acts as tumor suppressor (Yang *et al*, 2011; Tetreault *et al*, 2013). Considering that TP53 mutations are mutually exclusive with HPV infection in HNSCC, and that these two types of tumors (TP53mut/HPV− and

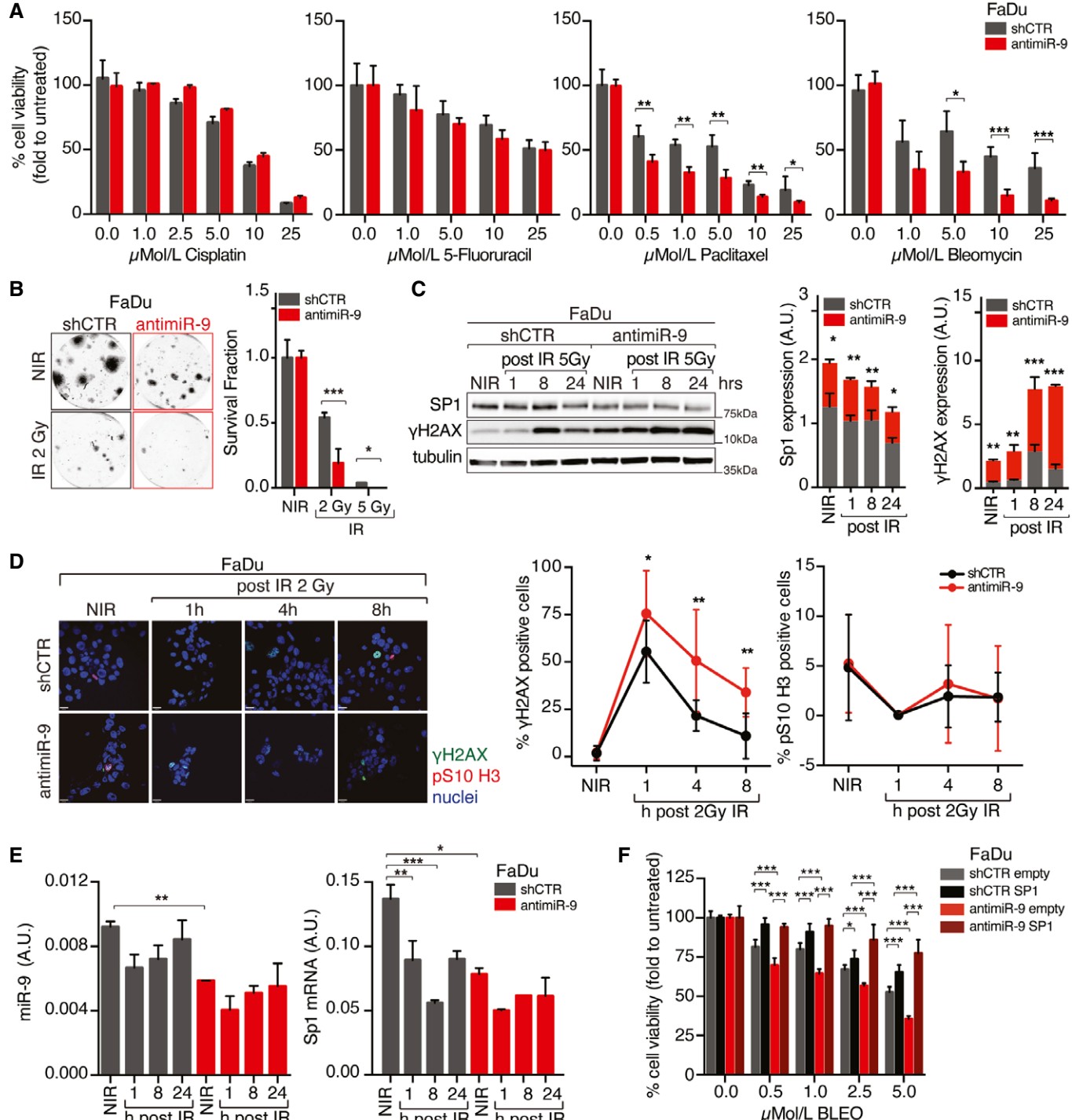

**Figure 5.**

**Figure 5.  miR-9 protects HNSCC cells from RT-induced cell death.**

A   Graph reporting cell viability of control (shCTR) and anti-miR-9 FaDu cells treated with increasing concentration of the indicated drugs for 72 h and analyzed using the MTS cell viability assay. Data represent the mean (± SD) of three independent experiments each performed in sextuplicate, and unpaired *t*-test was used to calculate the statistical significance per each dose.

B   Clonogenic assay of control (shCTR) and anti-miR-9 FaDu cells not irradiated (NIR) or treated with 2 or 5 Gy IR. On the left, typical images of cell clones are shown. On the right, the graph reports the percentage (± SD) of survived cells respect to not irradiate cells in three independent experiments performed in triplicate. Unpaired *t*-test was used to calculate the statistical significance per each dose.

C   Left, WB analyses of the indicated protein expression in control (shCTR) and anti-miR-9 FaDu cells not irradiated (NIR) or treated with 5 Gy IR and allowed to repair for the indicated hours (h). Tubulin was used as loading control. On the right, protein quantification analyses of KLF5 and Sp1 expression normalized on tubulin loading control. On the right, graphs report the quantification of the indicated proteins normalized on tubulin expression. Data are expressed as mean (± SD) of three independent experiments, and unpaired *t*-test was used to calculate the statistical significance at time point.

D   Left panel reports typical immunofluorescence images of control (shCTR) or anti-miR-9 FaDu cells, not irradiated (NIR) and analyzed 1, 4, or 8 h after 2 Gy IR (γH2AX green, pS10-H3 red, nuclei in blue). Graphs report the percentage of γH2AX (middle)- and pS10-H3 (right)-positive cells. Data represent the mean (± SD) of three independent experiments in which at least 10 randomly selected fields were evaluated. Unpaired *t*-test was used to calculate the statistical significance at each time point.

E   Graphs report the qRT–PCR analyses evaluating the expression of miR-9 (left) or SP1 (right) in cells treated as in (C). Data are the mean (± SD) of three independent experiment performed in duplicate. Two-way ANOVA with Sidak's multiple comparison test was used to calculate the statistical significance.

F   Graph reporting cell viability of control (shCTR) and anti-miR-9 FaDu cells, overexpressing or not SP1, and treated with increasing concentration of Bleomycin (BLEO) for 72 h and analyzed using the MTS cell viability assay. Data represent the mean (± SD) of three independent experiments each performed in sextuplicate. Two-way ANOVA with Sidak's multiple comparison test was used to verify the statistical significance.

Data information: In the figure, A.U. = arbitrary units; *P < 0.05; **P < 0.01; ***P < 0.001.
Source data are available online for this figure.

---

TP53wt/HPV+) might respond very differently to CTX+RT, the role of miR-9 and this double-faced behavior of KLF5 may represent very a relevant evidence that will need further exploration (Leemans *et al*, 2011). Two recent clinical trials have demonstrated that RT+CTX is not a feasible therapeutic opportunity in HPV-positive HNSCC patients, and thus, HPV infection cannot be considered a predictor of response to CTX (Gillison *et al*, 2019; Mehanna *et al*, 2019). These clinical evidences, along with our data, support the possibility that in a subset of TP53-mutated/HPV-negative HNSCC, the expression of miR-9 could be instrumental to promote tumor growth and the resistance to therapies. In this setting, we can foresee two scenarios: in miR-9 low tumors, the addition of CTX to RT, *via* blockage of the EGFR signaling pathway, will further decrease miR-9 expression, eventually contributing to the effectiveness of RT treatment; in miR-9 high tumors, the EGFR signaling pathway inhibition by CTX is not sufficient to dampen the miR-9/KLF5/Sp1 axis

and HNSCC will resist to treatments and eventually progress. In a TP53 wild-type context, it is conceivable that KLF5 down-regulation, either *via* miR-9 or by other mechanisms, will have only little effect on oncogenic pathways, such as the one of Sp1, as suggested by literature data (36,42). The significance and possible implications of mutant or wild-type TP53 in the pathway here described will be further investigated in the future. As for now, we have observed a different intrinsic sensitivity to gefitinib or Cetuximab of the different cell lines independently from their levels of miR-9 (e.g., comparing FaDu to SCC9 cells). Thus, it is possible that the endogenous levels of EGFR and/or EGFR ligands directly impact on the efficacy of EGFR blockade, especially in HPV-negative (mostly TP53mut) HNSCC (Wheeler *et al*, 2010; Huang *et al*, 2021).

Our data demonstrate that miR-9 transcription is rapidly induced by serum (1–2 h), supporting the possibility that immediate-early gene(s) are involved in the regulation of its promoter activity. c-Myc

---

**Figure 6.  miR-9 expression regulates and predicts the response to RT+Cetuximab combination therapy.**

A   Schema of the *in vivo* analyses of tumor growth in mice (n = 10/group) injected with FaDu cells. After tumor appearance, mice were injected intra-tumoral with high-titer viruses encoding for control or anti-miR-9 sequences. After two injections of virus, mice were treated with Cetuximab (IP injections) and RT (4 Gy dose) as indicated and then sacrificed 36 days after injections.

B   Graph reports the tumor volume of tumors described in A (n = 10 mice/group). Data represent the mean (± SD), and two-way ANOVA was used to verify the statistical significance.

C, D   Typical images of hematoxylin & eosin (H&E) (C) and Ki67 expression (Ki67) (D) evaluated by immunohistochemistry (IHC) in tumors explanted from mice treated as in (A, B).

E   Graph reports the percentage of Ki67-positive cells in tumors, as described in (D). Data are expressed as mean (± SD) of Ki67 percentage counted in five randomly selected fields per tumor, in at least four tumors per group. Two-way ANOVA with Sidak's multiple comparison test was used to verify the statistical significance.

F   TUNEL assay performed in tumors described in (C). On the left, typical immunofluorescence images (blue—nuclei, green—TUNEL). On the right, graph reports the percentage of TUNEL-positive cells in tumors as described in (C). Each dot represents a tumor. Data represent the mean (± SD), and two-way ANOVA with Sidak's multiple comparison test was used to verify the statistical significance.

G   Kaplan–Mayer curve evaluating the overall survival of HNSCC patients treated with RT+Cetuximab (CTX) combination included in the TCGA dataset, segregated on the expression of miR-9 in the primary tumor (low expression < 75,819 reads n = 8; high expression ≥ 75,819 reads n = 23). Number of evaluated samples (n) and P value are reported in the graph. Statistical significance was calculated with log-rank test.

H   Kaplan–Mayer curve evaluating the progression-free survival of HNSCC patients treated with RT+Cetuximab (CTX) combination at the CRO-Aviano National Cancer Institute and at the University Cattolica segregated based on miR-9 expression in primary tumors, defined as the expression in above (high expression n = 18) or below (low expression n = 17) the median expression, as defined by ddPCR. Hazard ratio (HR) and statistical significance were calculated with log-rank (Mantel–Cox) test and are reported in the graph.

Data information: In the figure, *P < 0.05; **P < 0.01; ***P < 0.001.

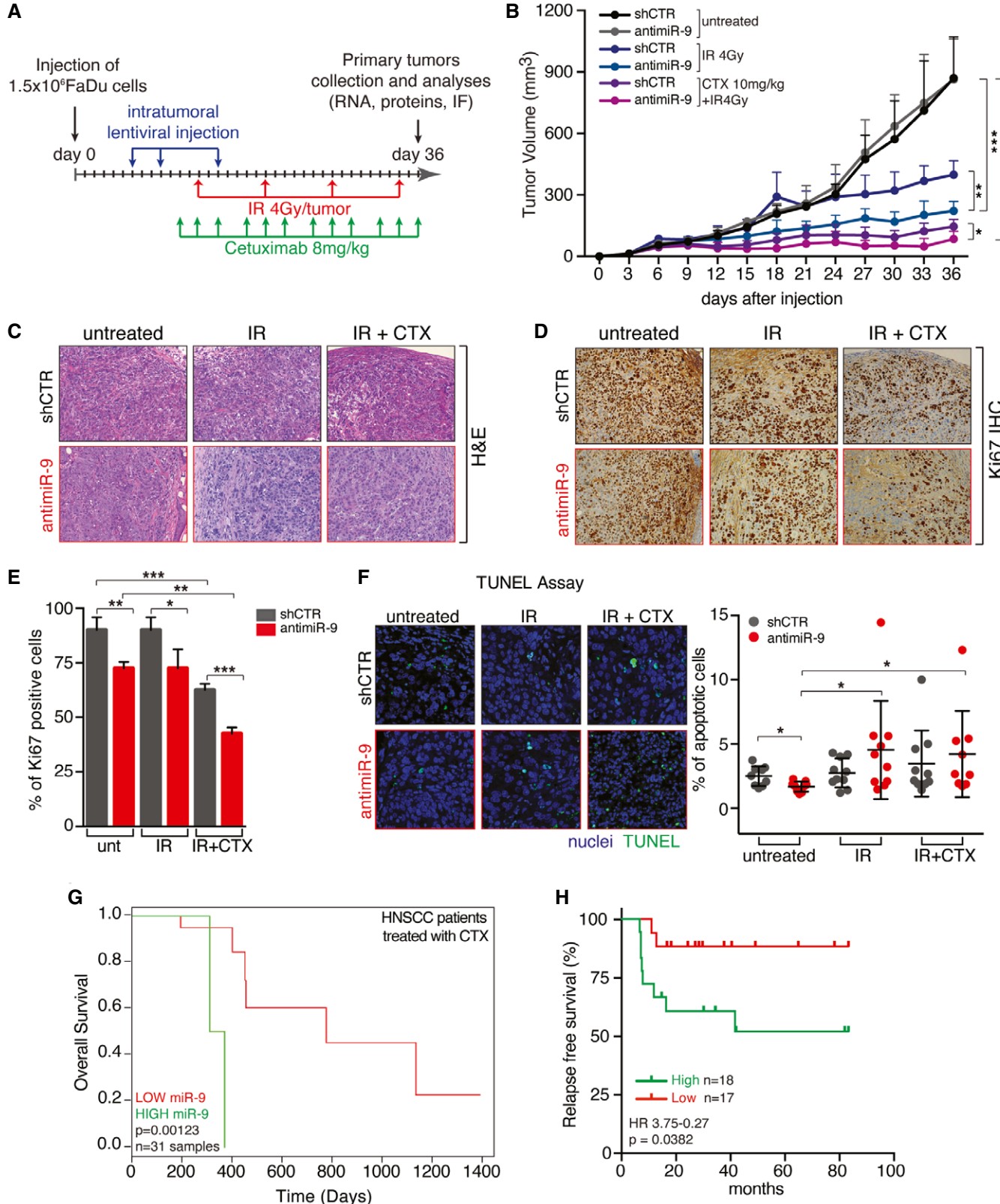

**Figure 6.**

is an immediate-early gene activated in response to several mitogenic stimuli, including EGF (Healy *et al,* 2013), and is activated and correlates with CTX resistance in a model of H-Ras-mutated HNSCC cells (Rampias *et al,* 2014). It has also been proposed that c-Myc is a positive regulator of miR-9 in breast cancer, where miR-9 acts as a positive modulator of EMT (Ma *et al,* 2010). Whether the same pathway is active in HNSCC cells is something that we will certainly address in the future. Indeed, we observed a mild correlation between EGFR and miR-9 expression in primary HNSCC. The weak nature of this correlation could be due to the fact that response to CTX in HPV-negative HNSCC is not entirely assessable by the level of EGFR activation, but needs a multiparametric assessment, as recently demonstrated (Huang *et al,* 2021).

The newly identified axis, linking EGFR activation to Sp1 expression *via* miR-9 and KLF5, represents a robust biomarker of CTX activity and can explain why only a subset of HNSCC patients benefits from the combined use of RT+CTX. On these bases, the evaluation of miR-9 expression in primary tumor biopsies, or, possibly, in the saliva of HNSCC patients (19), may represent a reliable biomarker to stratify patients and assign them to the most effective combination therapy. This personalized approach will not only allow to identify patients potentially more sensitive to RT+CTX, but also the ones potentially more sensitive to RT+platinum, eventually reaching a better outcome and sparing ineffective treatments and unwanted toxicities to many HNSCC patients.

# Materials and Methods

## Patient samples and study approval

Specimen collection. Specimens from primary HNSCC (Appendix Table S1) were collected from patients who underwent surgery at our Institution and at Santa Maria degli Angeli Hospital, Pordenone, Italy. HNSCC specimens were immediately frozen and stored at −80°C. Paraffin-embedded samples of radiotherapy plus Cetuximab-treated HNSCC patients were obtained from the Santa Maria degli Angeli Hospital (Pordenone, Italy), Isontina Hospitals (Monfalcone and Gorizia, Italy) and from the Fondazione Policlinico Universitario Agostino Gemelli Università Cattolica del Sacro Cuore (Rome, Italy) (Appendix Table S2). The study was approved by the Internal Review Board of the Centro di Riferimento Oncologico (CRO) of Aviano (#IRB-08/2013). A written informed consent was obtained from all patients included in this study, and the experiments conformed to the principles set out in the WMA Declaration of Helsinki and the Department of Health and Human Services Belmont Report.

Analysis of the TCGA dataset. miRNA-9 expression levels were obtained by The Cancer Genome Atlas (TCGA) from 31 HNSCC tissue samples from patients treated with Cetuximab and radiotherapy (cohort 1) and from 133 HNSCC tissue samples of patients treated with platinum and radiotherapy (cohort 2). For each cohort, the Kaplan–Meier method was performed to generate survival curves and the statistical significance of the difference between survival curves of high- vs. low-expression groups was evaluated using the log-rank test (Therneau & Grambsch, 2000). The cut-off point for the two groups was changed iteratively (*P*-values less than 0.01 were considered to be significant), and the cut-off that obtained the most significant *P* value was selected.

## Animal study approvals

Animal experimentation was approved our Institutional Animal Care and Use Committee (OPBA) and conducted strictly complying with internationally accepted guidelines (IACUC) for animal research and following the 3Rs' principles.

## Cell biology experiments

### Cell culture

FaDu, CAL27, UMSCC1, 293T17, and 239FT cells were cultured in Dulbecco modified Eagle medium (DMEM, Sigma) supplemented with 10% fetal bovine serum (FBS, Gibco). SCC9 cells were cultured in a 1:1 mixture of Dulbecco's modified Eagle's medium (Sigma) and Ham's F12 medium (Sigma) containing 1.2 g/l sodium bicarbonate (Sigma), 2.5 mM L-glutamine (Sigma), 15 mM HEPES (Sigma), and 0.5 mM sodium pyruvate (Sigma) supplemented with 400 ng/ml hydrocortisone (Sigma) and FBS 10%. HNBE cells were cultured in Airway Epithelial Cell Basal medium (ATCC) supplemented with Bronchial Epithelial Cell Growth Kit (ATCC). All the cells were routinely tested to exclude mycoplasma contamination (MycoAlertTM, Lonza) and authenticated by STR analysis in 2018, according to PowerPlex® 16 HS System (Promega) protocol and using GeneMapperTM software 5 (Thermo Fisher) to identify DNA STR profiles.

All *in vitro* studies were performed in triplicate, unless otherwise specified.

### Lentiviral transduction of HNSCC cells

FaDu, SCC9, Cal27, and UMSCC1 cells overexpressing or silenced for miR-9 expression were generated by lentiviral system as previously described (Citron *et al,* 2017). Briefly, lentiviruses expressing anti-miR-9-5p (MISSION® Lenti miRNA inhibitor human has-miR-9-5p, HLTUD0946) or expressing miR-9-5p (MISSION® Lentiviral miRNA transduction particles human has-miR-9-5p, HLMIR0946) were purchased by Sigma. Cells were transduced with anti-miR-9, miR-9 control lentivirus according to the manufacturer protocol and selected in 1.0 µg/ml puromycin.

Sp1 silenced cells were generated by lentiviral system, as described before (3). Briefly, 293FT cells were transfected with Gag-Pol and VSV-G (Invitrogen Lentivirus Production System) plus plasmid encoding the shRNA sequence against Sp1 (sh#3 TRCN00 00020448, sh#5 TRCN0000274153, Sigma) or against SASH1 (sh#2 TRCN0000162559, sh#4 TRCN0000165625) using calcium phosphate transfection kit (Promega), following manufacturer's protocol. After 48 and 72 h, conditional medium containing lentiviral particles was harvested and used to transduce target cells. Sp1 or GFP-miR-9-5p overexpressing cells and miR-9-sponge cells were generated by retroviral system, as described (Segatto *et al,* 2019). Briefly, 293T17 cells were transfected with pHIT465 and pHIT60 plus microRNA Precursor (MDH1-PGK-GFP-miR-9—Addgene #25036), Sp1 (RSV-Sp1—Addgene #12098), or miR-9 sponge sequence (pBABE miR-9 sponge—Addgene #25040) or Control Constructs by standard calcium phosphate transfection protocol. After 72 h, conditional medium containing retroviral particles was harvested and used to transduce target cells. Cell pools were selected in complete medium supplemented with 1.0–1.5 µg/ml puromycin.

FuGENE HD (Promega) transfection system was used following the manufacturer's instructions to overexpress different genes in HNSCC cells. Vectors used were as follows: pcDNA3-HA-KLF5 (Addgene #40904), pEGFP-SASH1 (kindly provided by Dr. KP Janssen), pcDNA3.2/V5 miR-9 (Addgene #26317), or control vector (pEGFP-C1 or pcDNA3.1, Clontech). Transfected cells were selected in complete medium supplemented with 500 μg/ml G418.

### Cell viability and IC50 drug calculation

For growth curve analyses, HNSCC cells were seeded into 96-well culture plates ($1-2 \times 10^3$ cells/well), and after 24 h, cell proliferation was measured with CellTiter 96 AQueous cell proliferation assay kit (Promega) every day for six consecutive days.

For kill curve analyses, HNSCC cells were seeded into 96-well culture plates ($2-4 \times 10^3$ cells/well), and after 24 h, cells were treated with increasing doses of drugs for 72 h (Citron et al, 2017). Cell viability was determined at the end of treatment using the Cell-Titer 96 AQueous Cell Proliferation Assay Kit (Promega).

### Sphere-forming assay

To establish primary spheres, cells were plated ($8 \times 10^3$) on poly-HEMA-coated dishes as single cell suspension in standard sphere medium containing phenol red-free DMEM/F12 (GIBCO), B27 supplement (50×, no vitamin A; Life Technologies) and recombinant epidermal growth factor (hEGF, 20 ng/ml; SIGMA). In a subset of experiments, mithramycin was added to the medium as indicated. After 8–10 days, primary spheres were counted and sphere area was measured with Volocity® software (PerkinElmer). To establish secondary spheres, primary spheres were collected, disaggregated in trypsin using 25-gauge needle fitted to a syringe. Cells were plated at the same seeding density of the primary generation. Sphere-forming efficiency (SFE%) was calculated using the following formula:

$$SFE\% = \text{No. spheres per well/No. of cells seeded per well} \times 100.$$

Sphere self-renewal was calculated as the ratio between the total number of secondary spheres divided for the total number of primary spheres.

### Luciferase assay

Luciferase assay was performed to validate miR-9 putative target sites on KLF5 3'UTR and KLF5 putative binding site on Sp1 promoter region as described (Nicolás et al, 2001; Citron et al, 2017). Briefly, the sequence surrounding putative miR-9-binding sites was amplified from FaDu cell genomic DNA using specific primers (Appendix Table S3). PCR products were cloned in the pGL3-basic vector (Promega) digested with XbaI (Promega), at the 3' of the luciferase gene, which is under the regulation of SV40 promoter. To generate mutant KLF5 3' UTR (mutant A and mutant B), side-directed mutagenesis of the WT (wild type) KLF5 3'UTR was performed using QuickChange II Site-Directed Mutagenesis Kit (Agilent #200523) according to the manufacturer's protocol.

Progressive deletion constructs of Sp1 promoter were amplified from FaDu cell genomic DNA using specific primers (Appendix Table S3). PCR products were cloned unidirectionally between the NheI and XhoI sites of the reporter luciferase vector pGL3 basic (Promega). These PCR fragments were generated using a common reverse primer and five different forward primers. The numbers indicated in the primer name correspond to the distance in nt from Sp1 ATG.

Luciferase assay to test miR-9 expression was performed using pMIR9 reporter vector (Addgene #25037) as described in (Ma et al, 2010). Briefly, CAL27 or FaDu cells were co-transfected with 500 ng of reporter constructs and 50 ng of pRL-TK vector (internal control) in 24-well plate using FuGENE® HD Transfection Reagent (Promega) according to manufacturer's recommendations. After transfection cell lysates were assayed for luciferase activity using the dual-luciferase reporter assay system (Promega). Values were normalized using Renilla luciferase.

### Clonogenic Assay

Cells were seeded into 6-well plates (500–2,000 cells/well depending on the cell lines) in complete medium and maintained at 37°C and 5% $CO_2$ for 10–15 days, refreshing the medium every 3–4 days. Colonies were then fixed and stained with 0.5 mg/ml crystal violet in 20% methanol. Colonies with more than approximately 50 cells were counted manually.

### Survival fraction assay

Irradiations were performed using Clinac 600 C (Varian Medical Systems, Palo Alto, CA) linear accelerator (LINAC) for external beam radiation therapy, at ambient oxygen concentrations and in cell adhesion conditions. Cell plates were positioned at the center of the radiation field of $40 \times 40 \text{ cm}^2$ size, with LINAC gantry at 180°, between two 5 cm layers of solid water. The dose delivered to the cell plates was 2 or 5 Gy at a dose rate of ~ 2.5 Gy/min, as calculated from measurements with radiochromic films in the same setup of irradiation.

Given the strong effect of miR-9 in mediate an increased cell survival, we plated shCTR or miR-9/anti-miR-9 cells accordingly to obtain a similar number of colonies in the untreated condition. This formula was used to calculate the correct number of cells to be plates is the following:

$$\begin{aligned} \text{No. of cell} = &\text{ No. of optimal counting colonies/} \\ &\text{plating efficiency in standard conditions/} \\ &\text{likelihood of predicted survival.} \end{aligned}$$

We then calculated the survival fraction as follow:

$$\begin{aligned} \text{Survival fraction} = &\text{ No. of clones in the IR condition/} \\ &\text{No. of clones in untreated condition.} \end{aligned}$$

Cells were seeded into 6-well plates or 60-mm dishes (two dilutions, in triplicate) and let adhere to the plates. Cells were then irradiated and maintained at 37°C and 5% $CO_2$ for 10–15 days, refreshing the medium every 3–4 days. Colonies were then fixed, stained, and counted as described in the Clonogenic Assay section. The survival fraction was expressed as the relative plating efficiencies of the irradiated cells to the control cells.

### Anchorage-independent soft agar assay

To evaluate the anchorage-independent cell growth, $1.5 \times 10^4$ FaDu cells stably transduced with control or anti-miR-9 were resuspended in 2 ml top agar medium (DMEM + 10% FBS, 0.4% low melting

agarose, SIGMA) and quickly overlaid on a previously gelified 0.6% bottom agar medium (DMEM + 10% FBS, 0.6% low melting agarose, SIGMA). The experiments were performed in six-well tissue culture plates, in triplicate. Fresh medium was added to the wells twice a week as a feeder layer. After three weeks, the number of colonies was counted in 10 randomly chosen fields, at 10× magnification.

### Random motility assay

Cells were seeded into 12-well plates ($2 \times 10^4$ cells/well) and allowed to adhere on the plates. In random motility assay, 8 cells/well were randomly selected and their x-y coordinates were obtained with 5-min step interval for 16 h. Images were collected using time-lapse microscopy (time-lapse AF6000LX workstation—Leica). Videos were generated assembling the images with the Volocity® software (PerkinElmer). Total distance and average cell speed were obtained for each cell using the measurement tool in ImageJ (NIH, USA).

### Migration assay

Cells were disaggregated as single cell suspension using trypsin, washed twice with PBS, and stained with PKH26 labeling solution according to manufacturer's protocol (Sigma). After washes, $5 \times 10^5$ cells were carefully resuspended in 100 µl of DMEM 0.1% BSA and seeded over a Transwell membrane (Corning) previously placed in a 24-well plate. Cells were allowed to settle for 10 min on the membrane; then, 400 µl of DMEM 10%FBS was added in the bottom of each well. The PKH26 fluorescence signal was acquired using Omega Microplate Reader (BMG Labtech) at the top and at the bottom of each well at different time points, and the fraction of migrated cells was calculated as following:

$$\% \, \text{migrated cells} = (\text{bottom fluorescence/top fluorescence}) * 100.$$

After the acquisition of the fluorescent signal, the Transwell membranes were removed from the 24-well plates and the cells, able to pass through the membranes, were allowed to attach for 8 h to capture representative images of the cells that efficiently migrated over the 2-h period.

### Evasion assay

Evasion assays were performed as previously described (Sonego et al, 2019). Briefly, $7.5 \times 10^3$ SASH1-modified CAL27 cells were included in Matrigel (Cultrex, BME) drops at a final concentration of 8 mg/ml (12 µl of matrix volume per drop). Matrigel was diluted in DMEM 1640 and 0.1% BSA. The drops, sufficiently spaced from one another, were dispensed in cell culture dishes and maintained for 1 h at 37°C upside down to jellify. Then, the dishes were turned up, and the drops were incubated in complete medium. The evasion ability was evaluated 10 days after inclusion by measuring the distance covered by crystal violet-stained cells exiting from the drops (five drops/cell lines per experiment). Images collected using a stereo microscope Leica M205FA.

### Reagents

Mithramycin A (MTA—Sp1 inhibitor) and gefitinib (EGFR small-molecule inhibitor) were purchased from Sigma and used for in vitro experiments. Bleomycin (BLEO), Cisplatin (CDDP),

paclitaxel (TAX), and 5-fluorouracil (5-FU) were purchased from TEVA Italia, and Cetuximab—Erbitux® was purchased from Merck.

## Molecular biology experiments

### RNA extraction and qRT–PCR analyses

Total RNA for qRT–PCR analyses was isolated from HNSCC primary tumors or cell cultures using TRIzol solution (Roche Applied Science Mannheim, Germany) according to manufacturer protocol. GentleMACS™ Dissociator (Miltenyi Biotec) was used to disrupt HNSCC primary tumors, and lysates were passed at least five times through a 23-gauge needle fitted to an RNase-free syringe. Total RNA was quantified using NanoDrop (Thermo Fisher Scientific Inc., USA).

The expression of miR-9-5p was analyzed using the TaqMan single-tube MicroRNA Assays (#000583 Thermo Fisher Scientific). All reagents, primers, and probes were obtained from Applied Biosystems. Reverse Transcriptase (RT) reactions and qRT–PCR were performed according to the manufacturer instructions (Applied Biosystems, Life Technologies). Normalization was performed on the U6 RNA (#001973 Thermo Fisher Scientific). All RT reactions were run in an T100 Thermal Cycler (Bio-Rad). Comparative qRT–PCR was performed in triplicate, including no-template controls. miR levels were quantified using the CFX384 (Bio-Rad). Relative expression was calculated using the comparative Ct method.

For gene expression analysis, RNA was retro-transcribed with GoScript Reverse Transcriptase to obtain cDNAs, according to provider's instruction (Promega). Absolute quantification of targets was evaluated by qRT–PCR, using SYBR Green dye-containing reaction buffer (SsoFast Master Mix 2×, Bio-Rad). The incorporation of the SYBR Green dye into the PCR products was monitored in real-time PCR, using the CFX384 Real-time PCR Detection System (Bio-Rad). Ct values were converted into attomoles, and normalized expression was evaluated by using SDHA and actin as housekeeping genes. Primers used in qRT–PCR are reported in Appendix Table S4.

### Droplet Digital PCR (ddPCR) assay

Total RNA for ddPCR analyses was isolated from FFPE HNSCC biopsies as described above. Total RNA was retro-transcribed and converted in cDNA using TaqMan-based technology. Briefly, this technology incorporates a target-specific stem-loop reverse-transcription primer which extends the length of mature microRNA (~ 22 bp) at its 3'. The resulting chimera, consisting of mature microRNA and the stem-loop primer, represents a template of a sufficient length to be analyzed with standard real time or ddPCR using TaqMan assays. Following cDNA synthesis, ddPCRs were prepared in a similar manner as qRT–PCRs. Briefly, ddPCR is composed by 1 ng RNA-equivalent cDNA, ddPCR™ supermix for probes (no dUTPs 2x—Bio-Rad) and the properly TaqMan probes for analyzing miR-9 and U6 (Applied Biosystems). The droplet generations were performed in a QX200 Droplet Generator (Bio-Rad) using Droplet Generation Oil for Probes (Bio-Rad) according to the manufacturer protocol. Thermocycling of the microfluidic emulsions was achieved in an Epp twin tec PCR plate 96 semi-skirted (Eppendorf) using T100 Thermocycler (Bio-Rad). PCR thermocycling was initiated with a 10 min "hot start" at 95°C to activate the polymerase, followed by 40 cycles at 94°C for 30 s and 59°C for 30 s, and the last step to inactivate the enzyme at 98°C 10 min. miR-9 and U6 absolute

quantification were achieved using QX200 Droplet Reader (Bio-Rad), and data were analyzed with QuantaSoft (Bio-Rad).

### Protein extraction, immunoprecipitation, and western blot analysis

For cellular protein lysates, cells were scraped on ice using cold Ripa lysis buffer (150 nM NaCl, 50 mM Tris–HCl pH 8, 1% Igepal, 0.5% sodium deoxycholate, 0.1% SDS) supplemented with a protease inhibitor cocktail (CompleteTM, Roche), 1 mM Na3VO4 (Sigma), 100 mM NaF (Sigma), and 1 mM DTT (Sigma).

Proteins were separated in 4–20% SDS–PAGE (Criterion Precast Gel, Bio-Rad) and transferred to nitrocellulose membranes (GE Healthcare). Membranes were blocked with 5% dried milk in TBS-0,1% Tween 20 or in Odyssey Blocking Buffer (LI-COR, Biosciences) and incubated at 4°C overnight with primary antibodies. The list of primary antibodies is provided in Appendix Table S5.

Membranes were washed in TBS-0,1% Tween 20 and incubated 1 h at RT with IR-conjugated (AlexaFluor680, Invitrogen or IRDye 800, Rockland) secondary antibodies for infrared detection (Odyssey Infrared Detection System, LI-COR) or with the appropriate horseradish peroxidase-conjugated secondary antibodies (GE Healthcare) for ECL detection (Clarity Western ECL Substrate, Bio-Rad). Band quantification was performed using the Odyssey v1.2 software (LI-COR) or the QuantiONE software (Bio-Rad Laboratories). The Re-Blot Plus Strong Solution (Millipore) was used to strip the membranes, when reblotting was needed.

### Chromatin immunoprecipitation (ChIP) assay

FaDu and CAL27 cells were treated with 1% formaldehyde, and chromatin was prepared via MNase enzymatic digestion according to the protocol. Chromatin immunoprecipitation (IP) was performed using SimpleChIP Enzymatic Chromatin IP kit—Magnetic Beads (#9003, Cell Signaling Technology). After IPs, DNA was purified and analyzed by qRT–PCR. Signals obtained from each IP are expressed as % of the total input chromatin. PCRs included the positive control histone H3, the negative control Normal Rabbit IgG, putative negative (Negative Control 1, 2, and 3), and putative positive controls (EPPK1 LAMC2, Epha, SERPINE1 INPP4B, and SOX17) of anti-KLF5 specificity and the 2% input chromatin DNA. The 2% of the amount of the chromatin is used for the input; thus, a 50-fold dilution factor has been taken into account $(Log2(50) = 5.64$ cycles), to calculate the adjusted input Ct value (100%), as follows:

$$Adjusted\ Input\ Ct = (Ct[Input] - [Log2(50)]$$

Then calculation of ΔCt was obtained, following the formula

$$\Delta Ct = Ct(Adjusted\ Input) - Ct(sample\ of\ interest).$$

Finally, % of Input was obtained for each sample: % of Input = $100 * 2^{(\Delta Ct)}$ representing the enrichment of antibody binding onto specific regions of Sp1 promoter. The primers used to amplify the indicated Sp1 promoter regions and positive and negative controls of the anti-KLF5 antibody derived from the literature are reported in Appendix Table S6.

### Immunofluorescence analysis

For immunofluorescence analyses on 2D cells, wells containing the cells seeded on round coverslips (Thermo Fisher Scientific) were fixed 10 min in 4% PFA, permeabilized 5 min in PBS 0.2% Triton, and blocked 1 h in PBS-5% NGS. Incubation with primary antibodies (pS10-H3 #06-570 and pS139-H2AX (γH2AX) #05-636 Millipore) was performed ON at 4°C. Incubations with primary antibodies were followed by 1 h at RT with secondary antibody (Alexa Fluor 633, 568 or 488, Invitrogen). Propidium iodide (3 μg/ml) containing RNaseA (100 μg/ml) or TO-PRO in PBS 1× was used to stain nuclei (respectively, 20 and 5 min at RT) and Alexa 546-conjugated phalloidin (Molecular Probes) for the actin staining (1 h at RT). Stained cells were observed using a confocal laser-scanning microscope (TSP2 or TSP8 Leica).

### Histological analysis and immunohistochemistry

Mouse xenograft samples were fixed in formalin (overnight at 4°C) and processed for standard paraffin embedding. Histological sections (5 μm thick) were made from the paraffin blocks, deparaffinated with xylene, and stained with hematoxylin and eosin (H&E), according to standard procedures. Images were collected with Leica microscope to measure the percentage of necrotic area in tumor section. Routine deparaffinization of all sections mounted on positive charge slides was carried out according to standard procedures, followed by rehydration through serial ethanol treatments. Slides were immersed in citrate buffer [0.01 M sodium citrate (pH 6.0)] and heated in a microwave oven at 600 W (three times for 5 min each) to enhance antigen retrieval. Endogenous peroxidase was blocked with 0.3% hydrogen peroxide in methanol for 30 min. Sections were immunostained with Ki67 or Sp1 antibodies (Appendix Table S5) according to manufacturer's protocol and standardized procedures.

### TUNEL assay

Detection of apoptosis was performed with TUNEL assay, using In Situ Cell Death Detection Kit, AP (Roche) on sections from FFPE HNSCC xenografts, according to the manufacturer's instructions. Apoptosis was calculated as the ratio of positive cells over the total number of cells per field. At least four different mice for each group and 10 fields/slice were analyzed.

## Xenograft growth in mouse and treatment

To evaluate the tumor growth and onset of HNSCC cells, primary tumors were established by subcutaneous injection of $1 \times 10^6$ FaDu (4 mice) or $5 \times 10^6$ CAL27 (3 mice) parental cells bilaterally in the flanks of female athymic nude mice (Charles River, 6 weeks old). Growth of primary tumors was monitored by measuring tumor width (W) and length (L) with a caliper three times per week and calculating tumor volume based on the formula: Tumor volume $(mm^3) = (W2 \times L)/2$.

To evaluate the role of miR-9 in tumor growth and onset, primary tumors were established by subcutaneous injection of 0.5–$2 \times 10^6$ FaDu (control and anti-miR-9) or CAL27 (control or miR-9) cells bilaterally in the flanks of female NSG mice (Experimental Radiation Oncology—MD Anderson Cancer Center) or by subcutaneous injection of $1 \times 10^6$ CAL27 (control and anti-miR-9) cells, bilaterally in the flanks of female NSG mice (Charles River, 6 weeks old). Growth of primary tumors was monitored as described above.

To evaluate the role of miR-9 in response to radiotherapy and/or Cetuximab, primary tumors were established by subcutaneous

**The paper explained**

**Problem**

In head and neck squamous cell carcinoma (HNSCC) patients, the combination of radiotherapy plus Cetuximab (RT+CTX) might be equally effective and less toxic than chemoradiotherapy. However, this is true only in a subset of patients and not in those with HPV-positive disease. Unfortunately, genomic analyses of HPV-negative HNSCC failed to detect specific biomarkers that would allow systematic identification of patients who might benefit most from RT+CTX treatment, suggesting that epigenetic regulations are involved.

**Results**

We have identified miR-9 as biomarker of poor response to RT+CTX in patients and better response to RT+platinum. At mechanistic level, EGFR activation in tumor cells induced miR-9, which then, by targeting KLF5, positively regulates the expression of Sp1, a transcription factor involved in the acquisition of stem-like properties and resistance to DNA damage, such as RT, in many cancer types. Noteworthy, Sp1 and miR-9 expression levels positively correlate in primary HNSCC.

**Impact**

Our study provides translational insights into the role of miR-9 expression in HNSCC tumors and suggests that the evaluation of miR-9 expression in primary tumor biopsies, and possibly in the saliva, might guide the physician in personalizing the therapy for HNSCC patients, selecting between RT+chemotherapy and RT+CTX, both already approved for this pathology.

injection of $1.5 \times 10^6$ FaDu parental cells (Day 0) bilaterally in the flanks of female athymic nude mice (Charles River, 6 weeks old). Growth of primary tumors was monitored as described above.

At days 6, 9, and 15 after cell injection, when tumors reached a volume of 15–20 mm$^3$, pre-anesthetized mice received an intratumor injection of high-titer lentiviral particles (control or anti-miR-9-5p, MISSION® Lenti miRNA inhibitor transduction particles, SIGMA). Mice were randomly divided into four groups according to experimental design (5 mice/group). Pre-anesthetized mice were subjected to radiotherapy 4 Gy/tumor at days 13, 20, 27, and 34 after injection. Radiotherapy was administered using Clinac 600 C (Varian Medical Systems, Palo Alto, CA) linear accelerator (LINAC) for external beam radiation therapy at 2 Gy/min dose. Vehicle or Cetuximab (8 mg/kg) was administered intraperitoneally two days before, the same day, and two days after radiotherapy. Unless tumor burden was incompatible with the well-being of the animals, mice were sacrificed at the end of the experiment and pathologically examined.

**Statistical analyses**

For the *in vivo* studies, no statistical methods were used to predetermine sample size. The experiments were not randomized except that mice were matched by age, and randomly assigned to specific treatment groups.

Tumor volumes were measured by the non-blinded investigator with the caliper, and no subjective methods were applied. Animals were randomized to the different treatment groups (i.e., vehicle, radiation, Cetuximab, or the combination), and no exclusion criteria were applied.

All graphs and statistical analyses were performed using PRISM (version 6, GraphPad, Inc.) and R, SAS Software 9.2. In all experiments, differences were considered significant when $P$ was < 0.05. Statistical analyses including Kaplan–Meyer survival analyses, paired and unpaired $t$-tests, Mann–Whitney unpaired $t$-test and Spearman correlation test, one-way and two-way ANOVA test, and Sidak's multiple comparison test were used as appropriate and as specified in each figure.

# Data availability

This study includes no data deposited in external repositories.

**Expanded View** for this article is available online.

## Acknowledgements

We are grateful to the patients who participated in this study. We thank Mrs Sara D'Andrea and all the staff of Radiotherapy Unit at CRO-Aviano for their valuable technical support. We thank Dr. V. Giuliani for providing the HNBE cell lines and Dr. KP Janssen for providing the SASH1 vector, Dr. D. Bartel for the pcDNA3.2/V5 miR-9 vector (through Addgene), Dr. B. Weinberg for the MDH1-PGK-GFP-miR-9 and the pBABE miR-9 sponge vectors (through Addgene), Dr. R. Tjian for the RSV-Sp1 vector (through Addgene), and Dr. J.T. Dong for the pcDNA3-HA-KLF5 vector (through Addgene). We thank all members of the SCICC lab of the Molecular Oncology Unit for supportive scientific discussions. This work was supported by CRO-Aviano Ricerca Corrente core grant (linea 1) of Ministero della Salute, by Regione Friuli Venezia Giulia (PERMID grant), by Ministero degli Affari Esteri e della Cooperazione Internazionale (PGR01036 grant), and by CRO-Aviano 5‰ grant to G. Baldassarre; by grant from Associazione Italiana Ricerca sul Cancro (AIRC) to B. Belletti (IG#20061); by fellowship from Associazione Italiana per la Ricerca sul Cancro (AIRC) to I. Segatto (#18171) and F. Citron (#20902); by Fondazione Umberto Veronesi (Global Fellowship 2019); and by Guido Berlucchi Foundation (Travel Grant 2019) to F. Citron. S. Srinivasan was supported by the CPRIT Research Training Grant (RP170067). F. Citron received funding from AIRC and from the European Union's Horizon 2020 research program under the Marie Sklodowska-Curie grant agreement no. 800924 and 23874.

## Author contributions

FC and GB involved in conception and designed the study; FC, IS, LM, IP, GLRV, AF, MA, and ICo contributed to development of methodology/investigated the study; GFr, GFa, FM, VL, VG, LB, GP, SSu, and Ave involved in acquisition of data (provided animals, acquired and managed patients, provided facilities, etc.); FC, ICa, SSr, IP, IS, BB, AVe, and GB analyzed and interpreted the data (e.g., statistical analysis, biostatistics, computational analysis); FC, IS, IP, BB, AVi, GFD, and GB wrote, reviewed, and/or revised the manuscript; FC, GFa, VL, and FM contributed to administrative, technical, or material support (i.e., reporting or organizing data, constructing databases).

## Conflict of interest

The authors declare that they have no conflict of interest.

## For more information

Online databases consulted in this study include the following:

- https://tcga-data.nci.nih.gov/tcga/tcgaDownload.jsp
- Refers to the TCGA website from which HNSCC data and the related literature can be obtained.

- https://cghub.ucsc.edu/
- Refers to the Santa Cruz University Genomic Institute website.
- https://www.atcc.org/~/media/C942A3363ED74FC0AAB46BE45D58ED1D.ashx
- Refers to the ATCC HNSCC cell lines panel website.
- https://depmap.org/portal/cell_line/ACH-000832?tab=mutation
- Refers to the Depmap website that describe the CAL27 HNSCC cell line.
- https://epd.epfl.ch/
- Refers to the Eukaryotic Promoter Database website.

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
