## [Review Process File · EMBO Molecular Medicine]

MiR-9 modulates and predicts the response to radiotherapy and EGFR-inhibition in HNSCC

Francesca Citron, Ilenia Segatto, Lorena Musco, Ilenia Pellarin, Gian Luca Rampioni Vinciguerra, Franchin Giovanni, Giuseppe Fanetti, Francesco Miccichè, Vittorio Giacomarra, Valentina Lupato, Andrea Favero, Isabella Concina, Sanjana Srinivasan, Michele Avanzo, Isabella Castiglioni, Luigi Barzan, Sandro Sulfaro, Gianluigi Petrone, Andrea Viale, Giulio Draetta, Andrea Vecchione, Barbara Belletti, and Gustavo Baldassarre

DOI: 10.15252/emmm.202012872

Corresponding author: Gustavo Baldassarre (gbaldassarre@cro.it)

Review Timeline:

Submission Date:	4th Jun 20
Editorial Decision:	10th Jul 20
Revision Received:	21st Feb 21
Editorial Decision:	19th Mar 21
Revision Received:	28th Apr 21
Accepted:	7th May 21

Editor: Zeljko Durdevic

Transaction Report:

Dear Dr. Baldassarre,

Thank you for the submission of your manuscript to EMBO Molecular Medicine. We have now heard back from the three referees who agreed to evaluate your manuscript. As you will see from the reports below, the referees acknowledge the interest of the study. However, they raise serious and partially overlapping concerns that should be addressed in a major revision of the present manuscript.

Overall it is clear that publication of the manuscript cannot be considered at this stage. I also note that addressing the reviewers concerns in full will be necessary for further considering the manuscript in our journal and this appears to require a lot of additional work and experimentation. I am unsure whether you will be able or willing to address those and return a revised manuscript within the six months deadline. On the other hand, given the potential interest of the findings, I would be willing to consider a revised manuscript with the understanding that the referee concerns must be fully addressed and that acceptance of the manuscript would entail a second round of review.

Please note that EMBO Molecular Medicine encourages a single round of revision only and therefore, acceptance or rejection of the manuscript will depend on the completeness of your responses included in the next, final version of the manuscript. For this reason, and to save you from any frustrations in the end, I would strongly advise against returning an incomplete revision and would also understand your decision if you chose to rather seek rapid publication elsewhere at this stage.

I look forward to receiving your revised manuscript.

Should you find that the requested revisions are not feasible within the constraints outlined here and choose, therefore, to submit your paper elsewhere, we would welcome a message to this effect.

Yours sincerely,

Zeljko Durdevic

***** Reviewer's comments *****

Referee #1 (Remarks for Author):

In this manuscript the authors identified a link between miR-9, KLF5, and SP1 that promotes resistance to radiotherapy in head and neck cancer, specifically in p53 mutated cells/tumors. The details of the p53 status effecting this response and a putative interaction of SP1 with TP53 is under further investigation for another study, which is appropriate. In this work, they further show that inhibition of EGFR signaling dampens miR-9 expression and that miR-9 levels are a potential biomarker for the response of targeting EGFR combined with RT treatment. (High miR-9 cannot be reduced enough in the presence of EGFRi, while if miR-9 is lower, inhibition of EGFR can reduce miR-9 enough such that KLF5 is upregulated and correspondingly SP1 is reduced leading to an effective therapeutic response). Based on the correlations in patients, it is conceivable that miR-9 expression could be a predictor of sensitivity to EGFR inhibition and radiotherapy. Overall the data presented are encouraging and are suggestive of this interesting link. The study is supported by a lot of experiments, many of which are fairly strong and well-controlled in cells, mice, and from human samples. There are however multiple concerns that warrant review by the authors. The most critical is inaccurate statistical analysis of most of the data presented. The manuscript would certainly be of interest to the readers of EMBO Molecular Medicine if these major concerns can be addressed.

- Q-RT-PCR data depicted in EV1 A was only conducted twice, and thus does not accurately support scientific rigor for this study.
- In Fig EV1 E, it is not clear what is being depicted. The heading for each graph is different (shCTR vs antimir-9) yet in each graph the legends suggest that both shCTR and anti-miR-9 are shown. Are both black lines shCTR as the legend suggests or are both lines in each graph from similar treatments as the heading suggests?
- Some discrepancy in tumor growth for the same cells in different assays. A little variability is expected; however, this is quite large. For example, in Fig2EVB at ~3.5 weeks (25 days) CAL27 shCTR cells grew in vivo to ~140mm³ while in Fig2EVH, tumors from CAL27 control cells at ~25 days were undetectable and after 8 weeks only reached ~40mm³. While half the number of cells were implanted in the later experiment, the discrepancy is larger than what would be expected. Even CAL27 cells overexpressing miR-9 in the later experiment took ~8 weeks to reach ~140mm³ - twice as long as cells overexpressing a control in the previous experiment, suggesting that miR-9 overexpression may not result in increased growth in vivo. It makes it difficult to analyze the data when growth of nearly the exact same cells varies from one experiment to another by this much, and that when miR-9 is overexpressed in one experiment the growth is slower than when it is not overexpressed in another experiment. At the very least this needs to be discussed, including why the number of injected cells varies from one experiment to the next.
- Why in some figures is a histone used as a loading control when the rest of the proteins are primarily cytoplasmic and not nuclear?
- For Fig2EV E, y-axis for the MTS assay should not be relative number of cells. There are multiple things that could cause the MTS signal to be altered other than cell number since this is a metabolic assay. The y-axis should accurately reflect the data. Correct as is accurately represented in Figure S1B.
- Because the authors did not verify a miR-9 targets site in KLF5 experimentally, there is still a hole in the study where miR-9 may not directly regulate KLF5. It is not essential to conduct this

additional experiment unless the authors are trying to portray that miR-9 directly regulates KLF5.

- Inflated citations by citing six manuscripts from the lab with the statement "All wet lab analyses were performed according to procedures commonly used in our lab". Specifically, Sonego 2019, Segatto 2014, Fabris 2016, and Dall'Acqua 2017, which are only cited in this one sentence and are not cited elsewhere in the manuscript.
- For Figure S1 (and in reagent list), the title indicating "anti-EGFR small inhibitors" needs clarification. What is meant by "small inhibitors"? Should it be small molecule inhibitors?
- For sphere forming efficiency calculation please be consistent and use either "N" as defined in supplemental methods or "n" as depicted in the figures (ie. Figure 1D,E).
- Some materials and methods are duplicated and are presented in both the main manuscript and in the supplemental without change. (ie. anchorage independent assay, random motility assay, etc.).
- A.U. in figures needs to be defined. (Arbitrary units?).
- It would be more useful to the reader to normalize data to the control such that an accurate change can be reflected. This appears to be done for some figures (Fig 2A 2C, where untreated is "normalized" to 1) but in other figures, data are not normalized to control (ie. Fig 2B where untreated is ~0.01). Fold change on the y-axis would be more useful to the reader than arbitrary units.
- The untreated samples should also have error bars associated with them, even if normalized, if done correctly. This is not evident for Fig 2A,2C,2E, etc. where it appears that data were normalized to untreated.
- For Figure 2G, ponceau staining of a membrane is not a good indicator of equal loading. A stronger loading control should be included, especially due to the changes observed in EGFR (lanes 1 and 2) and Y1068.
- Statistical calculations need to be adjusted appropriately. For example, in the majority of the data presented in Figure 2, the error bars are evaluating statistical significance across multiple samples. An unpaired t-test is not the appropriate test to use when more than two samples are being compared. Thus, the data in all figures where t-tests were used cannot be evaluated appropriately for significance. This needs to be rectified and all data reassessed using the correct statistical test and post-test. This is the case throughout the manuscript where repeatedly the incorrect statistical test is used.
- For droplet digital PCR methods clarity is needed. Line 3, states "...ddPCR supermix for probes no dUTPs 2x and the properly Taqman probes..." is this an error or were "no" dUTPs added? And what is meant by properly Taqman probes?
- Figure 4B is completely inaccurate based on the figure legend. There is no MTA treatment shown and loading control is Histone 3 and not Actin.
- Figure 1C - since the graph is not a curve (line graph connecting days) but instead are individual bar graphs, "growth curve" is not an appropriate term.

- For Figure 1, legend indicates that data are mean +/- SD for A, D-F. What is depicted in G?
- For Figure 3 G/H and 4D the reporter names are not consistent (G = -751, H = -788)
- For Figure 4E, legend states that graph is on the left, it is on the right. Also, a cartoon depicting where the primers lie on the SP1 promoter would be useful for the ChIP data. A negative and positive control are also missing from this experiment (a gene known to be bound by KLF5 and one that should not be bound). This is critical to assess the accuracy of the data and cleanliness of the IP. A conclusion cannot be rendered based on the data provided.
- For Figure 4E, there are no statistical marks (*, **, etc.) thus, analysis has not been conducted as indicated in the figure legend. Or was no significance noted? If this is the case, it should be indicated.
- Again, in Figure 5, the 0.0 treatments should still have error bars if performed in sextuplicate as indicated. The normalization needs to be done accurately to reflect the error. This needs to be looked over throughout the study.
- Ponceau S for a loading control for Figure 5E is not acceptable. Even with ponceuas S the first two lanes appear underloaded. An acceptable control needs to be included.
- In vivo data, while significant are only modestly different between control and anti-miR-9 treated tumors, especially in the CTX and IR treated group. The CTX and IR appear to be the best combination for treatment in this model.
- It is not clear how the authors quantified SP1 in the tumor samples to determine that SP1 was reduced in 8/10 tumors formed by anti-mir-9 FaDu cells (Fig EV5B). Based on the WB of SP1 in from these tumors and data quantified in Fig EV5C, it appears that SP1 was down in 6 tumors. Thus, the written results do not coincide with the data presented.

Referee #2 (Remarks for Author):

In this manuscript the authors describe the involvement of miR-9 in the HNSCC response to RT-CTX treatment. Firstly, they demonstrated that mir-9 controls the morphology, motility, sphere forming and self-renewal ability of HNSCC cell lines and that the activation of EGFR controls miR-9 expression in these cells. Notably, since the expression of EGFR is in turn modulated by miR-9 levels the authors also suggested a feedback regulation loop and the implication of miR-9 in the response to anti-EGFR treatment.

Searching for mir-9 targets they focused KLF5 (even though the manuscript lacks functional validation of this) and they identified a regulatory circuitry involving mir-9, KLF5 and SP1, transcriptionally repressed by KLF5. Since SP1 has been linked to altered DNA damage response and to the resistance to RT of HNSCC the authors suggested that high level of miR-9, by fueling the expression of SP1, might contribute to the aggressiveness of HNSCC cells. Indeed, they demonstrated that inhibition of miR-9 significantly improved the efficacy of RT and RT-CTX treatments in mice transplanted with FaDu cells. This effect was mainly attributed to EGFR/miR-9/SP1 axis even though in a previous study the same authors identified other two mir-9 targets with a potential role in HNSCC progression; nevertheless, the author did not consider the possibility

that more pathways might contribute to the observed effect.

However, it is worth noting that the relevance of miR-9 expression in predicting treatment response has been also verified in samples from HNSCC patients treated with RT+CTX .

Even though this is an attractive topic and the mir-9 regulated circuitries might add important insights for understanding the molecular bases of the resistance to treatments of HNSCC, the data provided are not in the shape making the manuscript suitable for publication: many controls are missing, many experiments needs replicates and quantification to support the authors conclusions and a more detailed description of the experimental tools is required.

In my opinion the manuscript needs a substantial revision before being considered for publication in EMBO Molecular Medicine.

Major concerns:

1. In Fig EV1B, P53 protein is shown with different MW in the two cell lines. Please explain this and describe in detail which p53 mutation occurs in the two cell lines.
2. Fig1B the reduction of N-Cadherin is clear while the significance of the upregulation of SASH1, KRT13 and ZO-1 is questionable without a quantification using biological replicates.
3. What is the rationale in using antimir-9 treated cells in experiments shown in Fig 2F and D.
4. In fig 2G and S1A in addition to total EGFR also pY1068 EGFR protein expression is shown, however this is never mentioned in the main text.
5. A comment on the different behavior of SCC9 and FaDu cells in fig S1B should be added. Moreover, a comparison between the levels of miR-9 in SCC9 respect to FaDu and CAL27cells is missing.
6. Why the author used different fragments of the SP1 promoter to perform the luciferase assay in figure EV3G? Moreover, they cited the work of Catalanotto et al 2016 for describing the nuclear activity of microRNAs on gene promoter but this is not the case of mir-9 that is supposed to be cytoplasmic. Moreover, the authors demonstrate that miR-9 regulate the activity of SP1 promoter in an indirect manner. Unless the authors show a nuclear localization of mir-9 in the cell lines used in this study there is no need to discuss about nuclear miRNA function.
7. The authors decided to focus on Klf5 as mir-9 target, however they did not perform any validation or cite any previous works reporting this (Yang et al. Cell Death Dis 2019, <https://doi.org/10.1038/s41419-019-2138-4>).
8. The effects of mir-9 modulation on KLF5 expression were assessed at RNA level for SSC9 while at protein level for CALC27 cells (Fig 4A-B), both the measurements have to be performed in both cell lines.
Western blot in fig 4C needs replicates and quantification to support the authors conclusion ("strong downregulation of SP1 mRNA and protein levels"). In control cells KLF5 is detected at higher molecular weight otherwise resulted not expressed at all. Could the authors make a comment on this?
Where are the KLF5 binding sites respect to the different portions of the SP1 promoter used in the luciferase assay (Fig 4D)? Please provide a schematic representation. Moreover, the authors should

also discuss why the same fragment of SP1 promoter leads to very different levels of luciferase transcriptional activation in FaDu vs CALC27 cells (Fig 4D vs EV3L), for instance -1612 is much more active in FaDu than in CALC27 while -281 has an opposite behavior.

Finally, since SP1 has been shown to be a target of miR-1, miR-133a, and miR-150 (Citron et al. 2017) the levels of these miRNAs should be measured in antimir-9 and/or mir-9 overexpressing FaDu and CALC27 cells in order to exclude that the regulation of SP1 expression is only due to KLF5 activity in these cells.

9. The chip experiment in Fig 4E lacks positive and negative controls. Moreover, the panel corresponding to the IgG amplification products has to be shown since the way to compute the ChiP data as fold over the IgG signal is allowed only if the IgG values between samples are equivalent. Expressing the IP values as % of input is preferred. Does "genomic DNA" stand for Input? Which percentage of In is used in the WB and in the PCR amplification?

10. The authors statement : CDDP, 5-FU and TAX "were equally effective in in both FaDu and SCC9 control and antimir-9 cells" is not totally correct since the endogenous miR-9 has a protective effect for TAX treatment in FaDu cells.

11. The quantification analysis shown in fig 5D has to be also performed for SCC9 cells in figure EV4D.

12. WBs in fig 5E and EV4F-G need replicates and quantification to support the author conclusions.

13. In figure EV5G the significance of the increased percentage of apoptosis in antimir-9 treated samples is computed respect to antimir-9 untreated ones. However, an increase of apoptosis is detected also in shCTR samples (not sure about the significance) upon treatments. In order to demonstrate that miR-9 has a role in the limitation of the efficacy of IR+CTX treatment the antimir-9 samples have to be compared with the shCTR treated ones. Is this difference still significant?

14. The authors also claim that "EGFR/miR-9/SP1 axis represents a major limit for the efficacy of RT+CTX.", what about the other two mir-9 targets SASH1 and KRT13 identified in a previous study by the same authors? As expected, both of them are present in the list in fig EV3I whit SASH1 being much more affected than KLF5, here shown to control SP1 expression. The author should make a comment on this and also touch upon this issue in the Discussion.

SASH1 and KRT13 are tumor suppressor genes with potential anti-EMT roles in HNSCC progression (Zeller et al., *Oncogene* 2003; Martini et al., *Int J Biochem Cell Biol* 2011; Yanagawa et al., *J Exp Clin Cancer Res* 2007). In order to demonstrate that EGFR/miR-9/SP1 axis is the major limits for the efficacy of the therapy a rescue experiment should be performed by forcing the expression of SP1 in antimir-9 treated FaDu parental cells.

15. In general, the experimental approaches used miss a detailed description: for instance the luciferase construct for testing the mir-9 promoter activation or the "two different systems" used to overexpress mir-9 in CAL27 cells. Moreover, it is not clear in all the quantitative RT-PCR performed how the levels of each target is computed (for instance which is the control mRNA or short RNA used to normalize the samples?). Is the $2^{-\Delta\Delta Ct}$ the value showed in the graphs?

Minor:

- Figure EV3 is nor cited properly: EV3G instead of EV3F, EV3H instead of G and so on....
- Check the reference format

Referee #3 (Remarks for Author):

The manuscript explores the mechanisms involved in a potential oncogenic function for mir9 in HNSCC. This study builds on a previous study from this group (PMCID: PMC7309652), where they identified a four-miRNA signature that classified HNSCC patients at high- or low-risk of recurrence. They showed that miR-9, targets SASH1 and KRT13 (confirmed here), whereas miR-1, miR-133, and miR-150, collectively target SP1 and TGF β pathways. In this study the authors now claim that mir9 also regulates SP1. This suggests a complex regulation of SP1 by multiple miRNAs in HNSCC. The experiments show that mir9 can induce the expression of SP1, through KLF5 suppression, which may contribute to promote tumorigenesis in HNSCC cells. The manuscript provides large amounts of data, but there is a lack of consistency and clarity in the data presentation that makes the paper hard to read: (1) there is little flow on the data description; (2) some experiments seem to contradict each other or just not support claims from previous figures; (3) it is unclear why they switch cell lines in some experiments; (4) in some experiments they show RNA expression of mir9 targets, in others they show protein, with no apparent reason; (5) different loading controls are used in different westerns, sometimes only ponceau staining is shown; (6) some comparative analyses might be statistically significant but the differences between groups are small and it is hard to predict whether those differences are biologically relevant which might question the significance of these studies; (7) the chemoradiation experiments (Fig 5) add little to the mechanistic studies of mir9 activities, and in fact, they add confusion because the conclusions are not well supported. (8) quantification of western blots and some staining is generally lacking. Below are specific comments:

1. They use two cell lines (Fadu and Cal27) in most experiments, but in some experiments one of these cell lines is substituted by SCC9. In most experiments they use one cell line for gain-of-function studies and another one for loss-of-function. They should use at least 2 cell lines for each gain-of-function and loss-of-function. SCC9 should have been included in the initial characterization of the cell lines shown in EV1.
2. The criteria used to count colonies in clonogenic assays should be described. In some experiments (e.g. Fig. EV4 C) the graphs don't seem to match the images shown.
3. Page 5 "In a panel of HNSCC-derived cell lines". Two cell lines cannot be considered a panel.
4. Page 5: "FaDu cells, associated with a mesenchymal phenotype". Mesenchymal markers are not shown. Same comment for the overexpression of mir9 in CAL27 (Fig EV2).
5. Mir9 expression is shown in two HNSCC cell lines, but not in normal oral epithelial cells. This is important to see the extent of upregulation of mir9 in these two cell lines.
6. Fig. 1. mir9 suppression leads to less than 50% reduction in mir9 expression. I am not sure if this is biologically significant. This is reflected in a moderate change in the mir9 targets KRT13 and SASH1. This comment affects many of the comparative analysis between controls and mir9 gain- or loss-of-function cells.
7. The expression of the proteins shown in Fig 1B needs to be quantified, after normalization to loading control.
8. Fig 1G. The data shown in this figure is contradicted by that shown in Fig 6A (untreated)
9. Fig 2. Western blot for EGFR phosphorylation in Cal27 cells with silenced mir9 should be shown, as nicely shown for mir9 overexpression in Fig 2G.
10. Fig 2H tries to make the point that a correlation between mir9 and EGFR exists in HNSCCs. However, the data indicates that this correlation is very weak ($r=0.18$).
11. The resistance to Gefitinib shown in Appendix Figure S1 is very modest for SCC9. I wonder why Cal27 was not used in these experiments to confirm the more robust effects observed in Fadu.

12. Gefitinib induces a rapid downregulation of mir9 on Fadu (Fig. 2C), as 2h after treatment approximately 75% of the mir9 expression is lost, a greater decrease in mir9 than that obtained with the anti-mir9 (Fig 1A). However, the cell viability is only affected when mir9 is suppressed with anti-mir9. This should be discussed and confirmed with other cell lines.
13. Fig. 3A lacks units.
14. Fig. 3C shCRT-control and anti-mir9-Control show about the same expression of SP1. According to Fig. 3B, should SP1 expression be lower in mir9-Control?
15. Fig. 3E-F. SP1 is known to control cell growth. The effects on SP1 suppression in cell growth in HNSCC cell lines should be shown. If cell growth is compromised by SP1 suppression, how would that affect the clonogenic and sphere formation ability of the cells?
16. The concentration of Mithramycin A used in different experiments should be indicated.
17. What is the control in the luciferase assays?
18. Fig. EV3F-L are not referenced correctly in the text.
19. Fig. EV3H. How was the 143-genes signature from Pavon et al generated? and why was it used to compare with the mir9 targets?
20. Fig. 4C. KLF5 induces, at the most, a modest reduction in SP1 expression. SP1 expression should be quantified after normalization to loading control. GRB2 is an unusual loading control.
21. Fig. 4E. The Chip'd DNA for control IgG is not shown.
22. Fig. 5. Drug/radiation response was analyzed at 72h. Considering that mir9 suppression results in slow cell growth, the response to the drug and radiation might be partially masked the cell growth effects. It would be cleaner to analyze the drug/radiation response at 24h. Cell density also affects the response. According to Fig 5D cell density of anti-mir9 cells was lower than that of control cells.
23. Fig 5C. According to the images of the cell dishes, mir9 overexpression doesn't seem to affect significantly the response to radiation. It seems that most of the effects conferred by mir9 are on cell growth.
24. Fig EV4D. Quantification should be shown.
25. Fig EV4E. Difference in gH2AX between control and mir9 is really small, might not be biologically significant. For pS10H3 it's concluded that radiation induces cell cycle arrest in control, but not in mir9 cells? What is the interpretation of the pS10H3 data in Fig 5D?
26. Fig 5E. The protein expression needs to be quantified, after normalization.
27. Fig 5E-F: "in irradiated FaDu cells, SP1 expression paralleled miR-9 levels showing a transient reduction of both protein and mRNA levels (Fig 5E-F)". This transient reduction of SP1 needs to be supported by the data; the protein was not quantified. And the reduction in RNA levels seems to persist at all time points (Fig. 5F).
28. Fig 5F: "Moreover, in antimiR-9 FaDu cells SP1 was significantly downregulated respect to controls and its expression did not recover up to 24 hours after irradiation (Fig 5E-F)". According to the data, it seems that radiation downregulates SP1 RNA in control but not in anti-mir9 cells.
29. Fig EV4F. Protein needs to be quantified.
30. Fig 6A. Fadu cells seem to be resistant to EGFR. In addition to validate this experiment with at least another cell line, EGFR expression and activation should be analyzed in control and anti-mir9 cells.
31. Page 11. "Interestingly, SP1 expression was reduced in tumors formed by antimiR-9 FaDu cells (in 8/10 mice)". This statement is not supported by the data; in the 10 pairs of tumors (control and anti-mir9), SP1 expression is lower in antimiR-9 in 4 of them, and higher or unchanged in 6 of them. The quantification graph shown in Fig EV5C does not seem to match the intensity of the SP1 bands for mice #2 and #4, even after normalization of GAPDH.
32. Fig 6C. The response to the combination in control and anti-mir9 cells might be statistically significant, but the difference is so small that is hard to see that this data is biologically significant.
33. Fig 6D needs to be quantified.
34. The number of patients in each group should indicated (Fig6 E-F, Fig EV5I).

Referee #1 (Remarks for Author):

In this manuscript the authors identified a link between miR-9, KLF5, and SP1 that promotes resistance to radiotherapy in head and neck cancer, specifically in p53 mutated cells/tumors. The details of the p53 status effecting this response and a putative interaction of SP1 with TP53 is under further investigation for another study, which is appropriate. In this work, they further show that inhibition of EGFR signaling dampens miR-9 expression and that miR-9 levels are a potential biomarker for the response of targeting EGFR combined with RT treatment. (High miR-9 cannot be reduced enough in the presence of EGFRi, while if miR-9 is lower, inhibition of EGFR can reduce miR-9 enough such that KLF5 is upregulated and correspondingly SP1 is reduced leading to an effective therapeutic response). Based on the correlations in patients, it is conceivable that miR-9 expression could be a predictor of sensitivity to EGFR inhibition and radiotherapy. Overall the data presented are encouraging and are suggestive of this interesting link. The study is supported by a lot of experiments, many of which are fairly strong and well-controlled in cells, mice, and from human samples. There are however multiple concerns that warrant review by the authors. The most critical is inaccurate statistical analysis of most of the data presented. The manuscript would certainly be of interest to the readers of EMBO Molecular Medicine if these major concerns can be addressed. We thank the Reviewer for Her/His appreciation of our work and for Her/His suggestions to improve it. As detailed below, we have now addressed all raised concerns, and we hope that She/He will find now our work acceptable for publication in EMBO Molecular Medicine.

- Q-RT-PCR data depicted in EV1 A was only conducted twice, and thus does not accurately support scientific rigor for this study.

We agree with the Reviewer and consequently we now provide a more compelling qRT-PCR analysis for miR-9 expression. The data presented in the new Appendix Fig. S1A are the results of technical duplicates performed in three independent experiments. We also broaden our analysis by including 4 different HNSCC cell lines, now used and characterized in the manuscript, and one normal control (NHBE – normal human epithelial cells) (Appendix Fig. S1A).

The new results have confirmed our previous data (Citron et al. 2017), showing that that CAL27 and UMSCC1 cells expressed the lowest level of miR-9, comparable with NHBE cells, whereas FaDu and SCC9 cells displayed the highest expression of miR-9.

- In Fig EV1 E, it is not clear what is being depicted. The heading for each graph is different (shCTR vs antimir-9) yet in each graph the legends suggest that both shCTR and anti-miR-9 are shown. Are both black lines shCTR as the legend suggests or are both lines in each graph from similar treatments as the heading suggests?

We apologize for this error in Fig EV1E labeling that has now been corrected and presented in the new Appendix Fig S1E. The left and the right panels showed the random motility of FaDu shCTR and anti-miR-9 cells, respectively. We also want to point out that this labeling error did not affect the quantification of cell motility plotted in the graphs and now reported in the new Appendix Fig S1F.

- Some discrepancy in tumor growth for the same cells in different assays. A little variability is expected; however, this is quite large. For example, in Fig2EVB at ~3.5 weeks (25 days) CAL27 shCTR cells grew *in vivo* to ~140mm³ while in Fig2EVH, tumors from CAL27 control cells at ~25 days were undetectable and after 8 weeks only reached ~40mm³. While half the number of cells were implanted in the later experiment, the discrepancy is larger than what would be expected. Even CAL27 cells overexpressing miR-9 in the later experiment took ~8 weeks to reach ~140mm³ - twice as long as cells overexpressing a control in the previous experiment, suggesting that miR-9 overexpression may not result in increased growth *in vivo*. It makes it difficult to analyze the data when growth of nearly the exact same cells varies from one experiment to another by this much, and that when miR-9 is overexpressed in one experiment the growth is slower than when it is not overexpressed in another experiment. At the very least this needs to be discussed, including why the number of injected cells varies from one experiment to the next.

We apologize for the incomplete information. The animals used in the previous Extended Fig. 2B were purchased from Charles River in Italy, while the animals in the experiment depicted in the former Extended Fig. 2H were purchased from Experimental Radiation Oncology at MD Anderson Cancer Center, USA. The different source of animals could have impacted on cancer cells growth more than expected.

Yet, we agree with Reviewer 1 that this discrepancy should be better clarified and we conducted a new *in vivo* experiment by injecting 0.5, 1 and 2 x10⁶ CAL27 and FaDu cells, modified for miR-9 expression, and evaluate the tumor onset and growth.

These new data, now included in the new Fig. 1G, show that the abrogation of miR-9 expression (anti-miR-9) in FaDu cells is sufficient to strongly delay the tumor onset. In CAL27 cells (new Fig EV1G) the overexpression of miR-9 anticipated the tumor onset respect to controls, an effect clearly visible when the lowest number of cells was injected (e.g. 0.5x10⁶ cells) that was mitigated when 1x10⁶ of cells were injected and disappeared when 2x10⁶ cells were injected. These data support the possibility that the different number of injected CAL27 cells could have impacted on the experiments showed in the previous version of the manuscript in Fig. EV2B and Fig. 2 E-H.

For clarity, in the new Figure EV1F, we now show only the growth of tumors formed by 0.5 million of CAL27. If the Editor and/or the Reviewers think that is necessary to add also the tumor growth

of a higher number of injected cells, we will be glad to add it in an updated version of the manuscript.

- Why in some figures is a histone used as a loading control when the rest of the proteins are primarily cytoplasmic and not nuclear?

We normally use more than one loading control in our experiments and compare them with the Ponceau staining of the membrane. We observed that Histone H3 expression was the more stable during our analyses. Yet we also used through the manuscript other loading controls including tubulin, actin or GAPDH when they reflected the total input intensity observed in the Ponceau staining.

- For Fig2EV E, y-axis for the MTS assay should not be relative number of cells. There are multiple things that could cause the MTS signal to be altered other than cell number since this is a metabolic assay. The y-axis should accurately reflect the data. Correct as is accurately represented in Figure S1B.

We have corrected the figures using the appropriate nomenclature.

- Because the authors did not verify a miR-9 targets site in KLF5 experimentally, there is still a hole in the study where miR-9 may not directly regulate KLF5. It is not essential to conduct this additional experiment unless the authors are trying to portray that miR-9 directly regulates KLF5. We thank Reviewer 1 for raising this central point. To investigate whether KLF5 could be a target of miR-9, we first interrogated two datasets (TargetScan - www.targetscan.org, and miRDB - www.mirdb.org) and found that the 3'UTR region of KLF5 contains two putative miR-9-binding sites. Next, we cloned the WT isoform of the KLF5-3'UTR in a luciferase reporter vector and starting from this we generated the mutant isoforms (mut. A, mut. B and mut. A+B), using a Site-Directed Mutagenesis Kit. We transfected CAL27 and FaDu cells modified for miR-9 expression and we analyzed the intensity of luciferase signal. The data reported in the new Fig 4D and Supplementary Fig EV3E demonstrated that in both cell lines miR-9 suppresses the LUC activity when the WT, the mut-A and the mut-B of KLF5 3'UTR were expressed, but not when the double mut A+B was expressed.

These data have been recently confirmed by others in HEK293 cells, using the 3'UTR of rat KLF5 and demonstrating that this regulation is conserved among the different species (PMID: 31787746).

- Inflated citations by citing six manuscripts from the lab with the statement "All wet lab analyses were performed according to procedures commonly used in our lab". Specifically, Sonogo 2019, Segatto 2014, Fabris 2016, and Dall'Acqua 2017, which are only cited in this one sentence and are not cited elsewhere in the manuscript.

The reason why we included all these citations is to sustain the expression "procedures commonly used in our lab". Yet, we understand this could appear as inflated citations and we agree to remove the not crucial citations from the manuscript.

- For Figure S1 (and in reagent list), the title indicating "anti-EGFR small inhibitors" needs clarification. What is meant by "small inhibitors"? Should it be small molecule inhibitors? Yes, we intended small molecule inhibitors, we apologize for the mistake. We have now included in the Supplementary methods the list of reagents used in cell biology experiments.

- For sphere forming efficiency calculation please be consistent and use either "N" as defined in supplemental methods or "n" as depicted in the figures (ie. Figure 1D,E).

We apologize that we have not explained well this point. Figure 1D and E did not report the sphere forming activity of FaDu cells but the number of colonies formed on plastic, in colony assay (Fig 1D) or in soft agar assay (Fig 1E). We have now noticed that in the previous version of the manuscript we omitted the methods for the colony assay that have been now included. We regret for this inaccuracy. The Sphere Forming Efficiency (SFE) is shown in Figure 1F, to quantify the potential cancer stem cells (CSC) phenotype of miR-9 modified cells. Sphere forming assay has intrinsically many technical issues to consider, not least the tendency of epithelial cells to form aggregates when plated in suspension that can lead to data misinterpretation and poor reproducibility. For this reason, we have used SFE, as it is the most appropriate way to express these results. SFE represents the percentage of spheres/number cell seeded and was calculated as reported in the method section, following a published protocol (PMID: 22914933).

- Some materials and methods are duplicated and are presented in both the main manuscript and in the supplemental without change. (ie. anchorage independent assay, random motility assay, etc.). We thank the Reviewer for highlighting this point and we have now removed the duplicated material and methods.

- A.U. in figures needs to be defined. (Arbitrary units?).

Yes, we used A.U. as the acronym of Arbitrary Unit, and we have now corrected the figure legends accordingly.

- It would be more useful to the reader to normalize data to the control such that an accurate change can be reflected. This appears to be done for some figures (Fig 2A-2C, where untreated is "normalized" to 1) but in other figures, data are not normalized to control (ie. Fig 2B where untreated is ~0.01). Fold change on the y-axis would be more useful to the reader than arbitrary units.

We thank Reviewer 1 for the suggestion and we have now modified the representation of the data. Now we have included the fold change for the graphs A-F, in the new Fig 2.

- The untreated samples should also have error bars associated with them, even if normalized, if done correctly. This is not evident for Fig 2A, 2C, 2E, etc. where it appears that data were normalized to untreated.

Yes, we apologize for this inaccuracy. We have now calculated and included the error bars of untreated controls.

- For Figure 2G, ponceau staining of a membrane is not a good indicator of equal loading. A stronger loading control should be included, especially due to the changes observed in EGFR (lanes 1 and 2) and Y1068.

To avoid misunderstanding and to give a clear and concise message, we decided to remove this panel from our manuscript, as we did not further investigate the activation of EGFR in response to different levels of miR-9. We only focused on the up-regulation of miR-9 in response to mitotic stimuli, including EGF.

- Statistical calculations need to be adjusted appropriately. For example, in the majority of the data presented in Figure 2, the error bars are evaluating statistical significance across multiple samples. An unpaired t-test is not the appropriate test to use when more than two samples are being compared. Thus, the data in all figures where t-tests were used cannot be evaluated appropriately for significance. This needs to be rectified and all data reassessed using the correct statistical test and post-test. This is the case throughout the manuscript where repeatedly the incorrect statistical test is used.

We thank the Reviewer for this comment and we modified the statistical significance calculated using the appropriate test.

- For droplet digital PCR methods clarity is needed. Line 3, states "...ddPCR supermix for probes no dUTPs 2x and the properly Taqman probes..." is this an error or were "no" dUTPs added? And what is meant by properly Taqman probes?

We extracted RNA using Trizol Reagent (Invitrogen) according to manufacturer's protocol. Total RNA was retro-transcribed and converted in cDNA using TaqMan-based technology. Briefly, this technology incorporates a target-specific stem-loop reverse-transcription primer with the aim to extend the length of mature microRNA (~ 22 bp) at its 3'. In this way, a chimera consisting of mature microRNA and the stem-loop primer represents a template of a sufficient length to be analyzed with standard real-time PCR using TaqMan assays. For this reason, we specifically chose a supermix for ddPCR that does not contain dUTPs (commercial name *Bio Rad ddPCR Supermix for Probes - No dUTP*), and no dUTPs were added since the template for ddPCR analyses was cDNA. Also, for each ddPCR analyses, we added to the ddPCR super mix the single TaqMan probe conjugated with a FAM fluorophore to analyze the expression of miR-9 or U6 independently. We have now included this explanation in the supplementary methods section.

- Figure 4B is completely inaccurate based on the figure legend. There is no MTA treatment shown and loading control is Histone 3 and not Actin.

We regret for this mistake and we have now corrected the figure legend.

- Figure 1C - since the graph is not a curve (line graph connecting days) but instead are individual bar graphs, "growth curve" is not an appropriate term.

We apologize for the inappropriate nomenclature. We have now corrected the manuscript and legend accordingly.

- For Figure 1, legend indicates that data are mean +/- SD for A, D-F. What is depicted in G? We thank the Reviewer for this question. In Figure G (new Fig 1H) we depicted the tumor volume expressed as mean value +/- SD. We have added this information in the figure legend.

- For Figure 3 G/H and 4D the reporter names are not consistent (G = -751, H = -788) We have now re-designed the figure, properly calculating for each construct the distance from the ATG. The error stemmed from the fact that multiple TSS (Transcription Start Sites) have been reported in the literature for SP1 promoter. We have accordingly modified the nomenclature used in the ChIP experiments (see new Fig 3I-J, Fig 4G-H and Fig EV3F-G).

- For Figure 4E, legend states that graph is on the left, it is on the right. Also, a cartoon depicting where the primers lie on the SP1 promoter would be useful for the ChIP data. A negative and positive control are also missing from this experiment (a gene known to be bound by KLF5 and one that should not be bound). This is critical to assess the accuracy of the data and cleanliness of the IP. A conclusion cannot be rendered based on the data provided.
- For Figure 4E, there are no statistical marks (*, **, etc.) thus, analysis has not been conducted as indicated in the figure legend. Or was no significance noted? If this is the case, it should be indicated.

We apologize for the inaccuracies. We have corrected the figure legends and provided a schematic representation of the Sp1 promoter, displaying the specific fragments cloned in the pGL3 luciferase vector and the ones used for ChIP experiments (new Fig 3I-J, Fig 4G-H and Fig EV3F-G).

We performed the ChIP assay in both FaDu (A) and CAL27 (B) cells including the negative (IgG) and the positive (Histone H3) controls. Below, the panels represent the KLF5 (left) and H3 (right) signals relative to the percentage of input of two independent ChIP analyses. Statistical significance is now reported and calculated with a two-way ANOVA test.

- Again, in Figure 5, the 0.0 treatments should still have error bars if performed in sextuplicate as indicated. The normalization needs to be done accurately to reflect the error. This needs to be looked over throughout the study.

We thank Reviewer 1 and we have now correctly applied the statistics and the normalization to all data throughout the study.

- Ponceau S for a loading control for Figure 5E is not acceptable. Even with ponceuas S the first tow lanes appear underloaded. An acceptable control needs to be included.

During our analyses we frequently observed that the expression of different controls was not paralleling the total input intensity. For this reason, in some experiments, we decided to include also ponceau staining (showing the total input) or H3 as loading control, as its expression results much

more stable during our analyses respect to tubulin and vinculin, also suggested by others (PMID: 23747530). Actually, the total input evaluation by intensity quantification of Ponceau Red stained lanes represents the most robust loading control, since it represents the entire lysate and is not affected by single protein fluctuations.

However, to meet Reviewer 1 request, we have now provided a new Western Blot analysis (new Fig 5C), using Tubulin as loading control and provide the quantification of blot. These results clearly confirm an earlier and more sustained increase in γ H2AX and a decreased expression of Sp1 in irradiated FaDu anti-miR-9 cells compared to controls.

- In vivo data, while significant are only modestly different between control and anti-miR-9 treated tumors, especially in the CTX and IR treated group. The CTX and IR appear to be the best combination for treatment in this model.

We agree with Reviewer 1 that the IR+CTX combination is the best combination for treatment in this model and that responses were only modest with the single agents. However, this approach was meant to highlight that only three intra-tumor injections of anti-miR-9 lentiviral particles were sufficient to improve the treatment efficacy. Since pre-silencing of miR-9 affected the *in vivo* growth of FaDu cells, it was otherwise difficult to properly evaluate specific miR-9 effects on treatment response. On the other side, this approach of intra-tumor injection of anti-miR-9 lentiviral particles lead to an incomplete miR-9 knock-down (see new Fig EV5E) and this of course also impacts on treatment response.

- It is not clear how the authors quantified SP1 in the tumor samples to determine that SP1 was reduced in 8/10 tumors formed by anti-mir-9 FaDu cells (Fig EV5B). Based on the WB of SP1 in from these tumors and data quantified in Fig EV5C, it appears that SP1 was down in 6 tumors. Thus, the written results do not coincide with the data presented.

In the previous version, we quantified the expression of SP1 and GAPDH, presenting the data as SP1 normalized expression relative to GAPDH in arbitrary units.

However, to better address this concern, we have now evaluated Sp1 expression using immunohistochemistry (IHC) that allowed us to more precisely evaluate Sp1 expression specifically in cancer cells. The Sp1 expression was then graded by a blinded pathologist based on nuclear staining intensity using four categories (0=negative, 1=very weak, 2=weak, 3=moderate, 4=strong). The data reported in the new Fig EV5D show that Sp1 expression was reduced in tumors formed by anti-miR-9 cells and further slightly reduced by CTX treatment.

Referee #2 (Remarks for Author):

In this manuscript the authors describe the involvement of miR-9 in the HNSCC response to RT-CTX treatment. Firstly, they demonstrated that mir-9 controls the morphology, motility, sphere forming and self-renewal ability of HNSCC cell lines and that the activation of EGFR controls miR-9 expression in these cells. Notably, since the expression of EGFR is in turn modulated by mir-9 levels the authors also suggested a feedback regulation loop and the implication of miR-9 in the response to anti-EGFR treatment.

Searching for mir-9 targets they focused KLF5 (even though the manuscript lacks functional validation of this) and they identified a regulatory circuitry involving mir-9, KLF5 and SP1, transcriptionally repressed by KLF5. Since SP1 has been linked to altered DNA damage response and to the resistance to RT of HNSCC the authors suggested that high level of miR-9, by fueling the expression of SP1, might contribute to the aggressiveness of HNSCC cells. Indeed, they demonstrated that inhibition of miR-9 significantly improved the efficacy of RT and RT-CTX treatments in mice transplanted with FaDu cells. This effect was mainly attributed to EGFR/mir-9/SP1 axis even though in a previous study the same authors identified other two mir-9 targets with a potential role in HNSCC progression; nevertheless, the author did not consider the possibility that more pathways might contribute to the observed effect.

However, it is worth noting that the relevance of miR-9 expression in predicting treatment response has been also verified in samples from HNSCC patients treated with RT+CTX. Even though this is an attractive topic and the mir-9 regulated circuitries might add important insights for understanding the molecular bases of the resistance to treatments of HNSCC, the data provided are not in the shape making the manuscript suitable for publication: many controls are missing, many experiments needs replicates and quantification to support the authors conclusions and a more detailed description of the experimental tools is required.

In my opinion the manuscript needs a substantial revision before being considered for publication in EMBO Molecular Medicine.

We thank the Reviewer for Her/His appreciation of our work and for Her/His suggestions to improve it. As detailed below, we have now addressed all raised concerns and we hope that She/He will find now our work acceptable for publication in EMBO Molecular Medicine.

Major concerns:

1. In Fig EV1B, P53 protein is shown with different MW in the two cell lines. Please explain this and describe in detail which p53 mutation occurs in the two cell lines.

We thank the Reviewer for this important request. In the revised manuscript, we have included the WB analysis of p53 in four different HNSCC cell lines, utilized and characterized in this work.

The table below reports the TP53 mutation of FaDu, CAL27 and SCC9 cell lines.

Cell line	Primary Tumor Source	Histology	Zygoty	Gene Sequence	Protein Sequence
FaDu	pharynx	SCC	heterozygous	c.743G>T	p.R248L
CAL27	tongue	SCC	homozygous	c. 578A>T	p.H193L
SCC9	tongue	SCC	homozygous	c.822_853del32	p.C275fs*20
UMSCC1	floor mouth	SCC	homozygous	Wild type	Splice-site mutant*

*UMSCC1 cells do not express endogenous p53 due to a splice-site mutation (hg19:chr17:7578370C > T).

We included in the new version of the manuscript the reference reporting the TP53 status of the used cell lines (PMID: 12884349.) We do not have an explanation for the apparent different MW of TP53 in FaDu and CAL27 cells that we imagine could be related to the different mutation they have.

2. Fig1B the reduction of N-Cadherin is clear while the significance of the upregulation of SASH1, KRT13 and ZO-1 is questionable without a quantification using biological replicates.

For the purpose of our manuscript, we decided to quantify SASH1 and KRT13, as a *bona fide* target of miR-9 (PMID: 28174235) and ZO-1 as a surrogate marker of the epithelial phenotype. Now, as requested, we have included the quantification of mean intensity of SASH1, KRT13 and ZO-1, analyzed in three biological replicates (see new in Fig. 1B).

To better complete the analysis, in the new version of the manuscript we also provide the expression and quantification of SASH1, KRT13 and ZO-1 in all the cell lines modified for miR-9 expression used in the manuscript (see new Fig 1B, Fig EV1B, Appendix Fig S2B, S2G and S2K).

3. What is the rationale in using antimir-9 treated cells in experiments shown in Fig 2F and D. We thank the Reviewer for this question. We observed that mitotic stimuli (i.e. FBS and EGF) promoted miR-9 transcription in a time dependent manner. Given the importance of EGFR in HNSCC tumors, we aimed to verify whether EGFR was specifically involved in the control of miR-9 expression and thus treated FaDu cells with EGFR inhibitor (i.e. Gefitinib or Cetuximab) and measured miR-9 expression. In this experiment we included antimir-9 FaDu cells as they are an

internal control expressing a low level of miR-9, which is biologically sufficient to induce a phenotype transition, and also to test whether EGFR blockade was able to further reduce the miR-9 level.

4. In fig 2G and S1A in addition to total EGFR also pY1068 EGFR protein expression is shown, however this is never mentioned in the main text.

We apologize for this inaccuracy and decided to remove these data from the revised version of our manuscript, focusing on the up-regulation of miR-9 in response to mitotic stimuli, including EGF.

5. A comment on the different behavior of SCC9 and FaDu cells in fig S1B should be added.

Our data and the current literature (PMID: 20551942 and PMID: 33417831) support the possibility that the endogenous levels of EGFR might impact on the efficacy of EGFR blockade. Indeed, we observed that SCC9 cells that express lower levels of EGFR respect to FaDu cells were intrinsically more resistant to EGFR blockade and that therefore the efficacy of antimiR-9 in increasing their sensitivity to Gefitinib and/or Cetuximab was visible at higher drugs doses. Now, we have added this comment in the new version of the manuscript (see page 17 lanes 2-5).

Below, for the Reviewer only, we report the levels of EGFR in the two cell lines analyzed by WB.

Moreover, a comparison between the levels of miR-9 in SCC9 respect to FaDu and CAL27 cells is missing.

We did not include these data only because they were already published in our previous work (PMID: 28174235). However, we agree with the Reviewer that it would be easier for the readers to have these data available also in this manuscript and therefore we have now included a new qRT-PCR analysis testing miR-9 expression in all HNSCC cell lines used in this manuscript. The results confirmed our previous data showing that CAL27 and UMSCC1 cells are expressing the lowest level of miR-9. Interestingly, miR-9 expression is also very low in normal epithelial NHBE cells (see new Fig S1A).

6. Why the author used different fragments of the SP1 promoter to perform the luciferase assay in figure EV3G? Moreover, they cited the work of Catalanotto et al 2016 for describing the nuclear activity of microRNAs on gene promoter but this is not the case of mir-9 that is supposed to be

cytoplasmic. Moreover, the authors demonstrate that miR-9 regulate the activity of SP1 promoter in an indirect manner. Unless the authors show a nuclear localization of mir-9 in the cell lines used in this study there is no need to discuss about nuclear miRNA function.

We thank the Reviewer for this comment that allow us to better explain how we decided to study the positive regulation of Sp1 by miR-9.

To respond to the first part of the question, we used different fragments of SP1 promoter to exclude/confirm a nuclear role for miR-9 and eventually map a possible binding region. However, since our experiments excluded a possible miR-9 nuclear role, these analyses are redundant. On the other side, this approach confirmed that the down-modulation of SP1 promoter activity in anti-miR-9 cells is specific because it is observed with all constructs but the shorter one, which does not have any significant activity (i.e. -146/+20).

For a better clarity, we now only describe the results without mentioning the possible nuclear activities of microRNAs (see new manuscript page 10 lane 1-6).

7. The authors decided to focus on Klf5 as mir-9 target, however they did not perform any validation or cite any previous works reporting this (Yang et al. Cell Death Dis 2019, <https://doi.org/10.1038/s41419-019-2138-4>).

We thank the Reviewer for raising this important point and for providing this relevant reference that is now cited in the text. However, that study refers to rat-KLF5 3'-UTR, tested in HEK293 cells. We have interrogated two datasets (TargetScan - www.targetscan.org, and miRDB - www.mirdb.org) and found that also the human 3'UTR region of KLF5 contains the same putative miR-9-binding sites reported in Yang et al. We cloned the WT isoform and three mutant isoforms (mut. A, mut. B and mut. A+B) of the KLF5-3'UTR in a luciferase reporter vector and transfected CAL27 and FaDu cells modified for miR-9 expression. The data, now shown in the new Fig 4C-D and Fig EV3E, demonstrate that miR-9 suppress the Luc activity in both cell lines when the WT, the mut A and the mut B were expressed, but not with the mut A+B double mutant.

8. The effects of mir-9 modulation on KLF5 expression were assessed at RNA level for SSC9 while at protein level for CALC27 cells (Fig 4A-B), both the measurements have to be performed in both cell lines.

As requested, we have now provided the Western Blot analysis for KLF5 expression in SCC9 shCTR and anti-miR-9 cells and the qRT-PCR for KLF5 expression in CAL27 shCTR and miR-9 (see new Fig EV3B-D).

Western blot in fig 4C needs replicates and quantification to support the authors conclusion ("strong downregulation of SP1 mRNA and protein levels").

As requested, we have now quantified the blot in the biological triplicates and these data are reported under the blot in the new Fig 4F.

In control cells KLF5 is detected at higher molecular weight otherwise resulted not expressed at all. Could the authors make a comment on this?

The vector used to overexpress KLF5 was obtained by the Addgene consortium (#40904 pcDNA3-HA-KLF5-his) and described in PMID: 21542805 and it has a HA-tag at the 5' and a HIS-tag at the 3' that explain the slightly higher molecular weight in western blot analyses. For clarity, we now provided the WB analyses using the HA antibody (new Fig 4F).

Where are the KLF5 binding sites respect to the different portions of the SP1 promoter used in the luciferase assay (Fig 4D)? Please provide a schematic representation.

We have now provided a schematic representation of the SP1 promoter with the specific fragments cloned in the pGL3 luciferase vector and the ones used for ChIP experiments. We have also indicated the regions on SP1 promoter predicted to be bound by KLF5 from our *in silico* analyses (see new Fig 3I).

Moreover, the authors should also discuss why the same fragment of SP1 promoter leads to very different levels of luciferase transcriptional activation in FaDu vs CALC27 cells (Fig 4D vs EV3L), for instance -1612 is much more active in FaDu than in CALC27 while -281 has an opposite behavior.

Thank you for this question. Regulation of gene transcription is a complex mechanism dependent on the formation of transcriptional units formed on the DNA upon the binding of the transcription factor to the promoter. These complexities are further enhanced by several post-translational modifications (*i.e.* acetylation, methylation, etc) that can greatly modify the activity of the transcriptional units. Therefore, it is possible that different transcriptional units are loaded on the different region of the SP1 promoter upon the binding of KLF5 and that these units are partially distinct in CAL27 and in FaDu cells. Alternatively, it is also possible that one (or more) transcriptional inhibitor(s), binding the SP1 promoter between -433 and -1612 bp from TSS, are expressed in CAL27 cells but are not present in FaDu cells. Of course, the full understanding of these transcriptional units goes far beyond the scope of this work, while the novel and important

data provided here is that, in both cell lines, KLF5 has a similar inhibitory activity on SP1 transcription and binds the same regions (see ChIP data in the new Fig 4H and Fig EV3G).

Finally, since SP1 has been shown to be a target of miR-1, miR-133a, and miR-150 (Citron et al. 2017) the levels of these miRNAs should be measured in antimir-9 and/or mir-9 overexpressing FaDu and CALC27 cells in order to exclude that the regulation of SP1 expression is only due to KLF5 activity in these cells.

We thank the Reviewer for this suggestion. To address this question, we analyzed the expression of miR-1, miR-133a, and miR-150 by qRT-PCR in FaDu control and antimir-9 cells, and also in CALC27 cells stably expressing antimir-9, miR-9 or control vector. A positive control for each miR in each cell line was included (i.e transfection with miR-1, miR-133a, miR-150 oligomiR).

As shown in the panels below, we confirmed our previous findings, that both FaDu and CALC27 display very low level, barely detectable, of each microRNAs. Moreover, neither the inhibition or the overexpression of miR-9 significantly altered the expression of these three microRNAs.

Based on these results, we decided to not add these data to the manuscript.

9. The chip experiment in Fig 4E lacks positive and negative controls. Moreover, the panel corresponding to the IgG amplification products has to be shown since the way to compute the ChIP data as fold over the IgG signal is allowed only if the IgG values between samples are equivalent. Expressing the IP values as % of input is preferred. Does "genomic DNA" stand for Input? Which percentage of In is used in the WB and in the PCR amplification?

We thank the Reviewer for this comment. Now, we have better explained how the former Fig. 4G was assembled to include all relevant controls (new Fig 4H). We show the WB analysis highlighting the specificity of the antibody used for the IP of KLF5 in ChIP and an agarose gel showing that all primers designed for the amplification of Sp1 promoter in the ChIP analyses specifically amplified both the DNA eluted from the ChIP assay and the genomic DNA of FaDu cells (no smears and a single band present on the gel after 40 cycles of PCR).

As requested by the Reviewer, we have graphed the ChIP data as signal relative to input and we also have included new ChIP analyses performed in CAL27 cells (new Fig 4H and Fig EV3G). The panels below report the ChIP experiments performed in FaDu (A) and CAL27 (B) cells. Negative (IgG) and positive (Histone H3, provided by the commercial kit) controls are now included and the data are presented as % of input. We also calculated the folds over control (IgG) and the results are extremely comparable (see panel C and D of the figure below, not included in the manuscript). The new Method section reports in detail how ChIP was performed and how % of input was calculated. Statistical significance is reported and calculated with a two-way ANOVA test. In the manuscript we have omitted to include the Histone H3 ChIP that, in our opinion, represents a redundant control. However, if the Editor and/or the Reviewer think that it would be necessary to have this control in the manuscript we are willing to add it.

10. The authors statement: CDDP, 5-FU and TAX "were equally effective in in both FaDu and SCC9 control and anti-miR-9 cells" is not totally correct since the endogenous miR-9 has a protective effect for TAX treatment in FaDu cells.

We agree with the Reviewer and now better discuss these results in the text (see page 11 lane 17-22). We have also broadened our analyses by treating CAL27 cells overexpressing or not miR-9 with CDDP, 5-FU, TAX and Bleo (see new Appendix Fig S6A) and these new data fully support the previous observations.

11. The quantification analysis shown in fig 5D has to be also performed for SCC9 cells in figure EV4D.

We agree with the Reviewer on this point. To have a realistic quantification of these IF analysis is necessary to evaluate and quantify multiple cells in different randomly selected fields for which we did not have the proper acquisition. We thus decided to remove the IF analysis performed in SCC9 and add instead new analysis (by WB) on FaDu, SCC9, CAL27 and UMSCC1 modified for miR-9 expression and irradiated with 2 or 5Gy, depending on the cell line sensitivity. As shown in Fig 5C, EV4C and Appendix S6C, WB analyses showed an increased activation of γ H2AX, a *bona fide* marker of DNA damage, in anti-miR-9 FaDu and SCC9 compared to controls. On the contrary, miR-9 overexpression in CAL27 and UMSCC1 cells led to a decreased level of γ H2AX.

12. WBs in fig 5E and EV4F-G need replicates and quantification to support the author conclusions.

As mentioned above, we have now performed multiple western blots on different cell lines (see new Fig 5C, EV4C and Appendix S6C) to more robustly support our conclusions. Quantification of the blots is provided, as requested.

13. In figure EV5G the significance of the increased percentage of apoptosis in anti-miR-9 treated samples is computed respect to anti-miR-9 untreated ones. However, an increase of apoptosis is detected also in shCTR samples (not sure about the significance) upon treatments. In order to demonstrate that miR-9 has a role in the limitation of the efficacy of IR+CTX treatment the anti-miR-9 samples have to be compared with the shCTR treated ones. Is this difference still significant?

Yes, the difference in apoptotic cells between shCTR and anti-miR-9 samples is not significant, although there was a trend toward it (using the two-way ANOVA test). We believe that the number of analyzed fields could have had an impact on the statistical significance. Unfortunately, we could not increase the number of analyzed fields since the old IF staining in these months partially lost their fluorescence and we did not have multiple field acquisitions. Conversely we could increase the number of analyzed fields/tumors in the IHC staining for Ki67 obtaining a stronger statistical significance (see new figure 6D-E).

Yet, we partially disagree with the Reviewer on the interpretation of these results. We think that the *in vivo* experiment should be considered as a whole and not focusing on a single panel.

We think that data from this experiment clearly demonstrate that:

- a) Only three intra-tumor injections of anti-miR-9 lentiviral particles are able to significantly impact on tumor growth (new Fig 6B), both in the case of IR alone and in the case of IR+CTX.
- b) This effect is specific for the treatments because the intra-tumor injection of anti-miR-9 alone has no effect on tumor growth, although it slightly affects cell proliferation (see Ki67 staining in new Fig 6D-E).
- c) IR alone does not alter the number of Ki67 positive cells, both in shCTR and anti-miR-9 tumors, compared to the untreated condition. However, IR alone significantly increases the percentage of apoptotic cells only in anti-miR-9 tumors, further confirming a synergistic cooperation between miR-9 inhibition and radiation.
- d) Strikingly, both in shCTR and anti-miR-9 groups, the administration of IR+CTX impacts on cell proliferation and apoptosis. The combination treatment is more effective on the proliferation and a slightly higher effect on apoptosis of anti-miR-9 tumors (new Fig 6).
- e) Finally, anti-miR-9 tumors have a higher necrotic area respect to controls, both after IR and IR-CTX treatments.

Based on these observations, we can conclude that miR-9 plays a role in limiting the efficacy of both IR and IR+CTX treatment, in the used mouse model.

14. The authors also claim that "EGFR/miR-9/SP1 axis represents a major limit for the efficacy of RT+CTX.", what about the other two miR-9 targets SASH1 and KRT13 identified in a previous study by the same authors? As expected, both of them are present in the list in fig EV3I whit SASH1 being much more affected than KLF5, here shown to control SP1 expression. The author should make a comment on this and also touch upon this issue in the Discussion. SASH1 and KRT13 are tumor suppressor genes with potential anti-EMT roles in HNSCC progression (Zeller et al., *Oncogene* 2003; Martini et al., *Int J Biochem Cell Biol* 2011; Yanagawa et al., *J Exp Clin Cancer Res* 2007).

We thank the Reviewer for this observation. Indeed, we have collected a huge amount of negative data on these miR-9 targets. Based on our previous results (Citron et al. 2017), SASH1 was our first choice to possibly explain the role of miR-9 in treatment response. However, all the results collected in these years to test this hypothesis on FaDu and CAL27 cells modified for SASH1 expression (now collected in the new Appendix Fig S4) failed to show any role for SASH1 in this setting. The overexpression of SASH1, whose vector was gently provided by dr. KP Janssen, in both FaDu and CAL27 cells did not alter the proliferation, colony formation and sphere generation

abilities. We also tested the motility of CAL27 cells, but SASH1 did not alter their migration, although it has to be considered that these cells display a very low migratory behavior.

We then silenced SASH1 expression in CAL27 cells by commercial lentiviral vectors, SIGMA-MERCK, observing only a slightly increase in the proliferation, not accompanied by changes in colony formation or migration.

These results, now included in new Appendix Fig S4, may be in line with literature data showing SASH1 expression is a prognostic marker in lung adenocarcinoma but not lung squamous carcinoma (PMID: 33122723).

In order to demonstrate that EGFR/miR-9/SP1 axis is the major limits for the efficacy of the therapy a rescue experiment should be performed by forcing the expression of SP1 in antimir-9 treated FaDu parental cells.

In order to address this question, we overexpressed SP1 in shCTR and antimir-9 CAL27 or FaDu cells. We observed that SP1 overexpression is sufficient to completely rescue the sensitivity of antimir-9 cells to both anti-EGFR blockade and Bleomycin. Interestingly, SP1 overexpression in shCTR cells strongly increased the resistance to bleomycin and did not alter the response to anti-EGFR treatment. These new data are now included in the new Fig 3H, Fig 5F, Fig EV2D and Appendix Fig S6E.

15. In general, the experimental approaches used miss a detailed description: for instance the luciferase construct for testing the mir-9 promoter activation or the "two different systems" used to overexpress mir-9 in CAL27 cells. Moreover, it is not clear in all the quantitative RT-PCR performed how the levels of each target is computed (for instance which is the control mRNA or short RNA used to normalize the samples?). Is the $2^{-\Delta ct}$ the value showed in the graphs? We apologize for this lack of details. In the revised version of manuscript, we now provide a detailed description of the methodologies used to conduct the experiments and for normalization. Specifically, by "two systems", we meant that the overexpression/silencing of miR-9 was achieved using different vectors, i.e. transfecting or transducing FaDu and CAL27 cells.

Minor:

- Figure EV3 is not cited properly: EV3G instead of EV3F, EV3H instead of G and so on....
- Check the reference format

Thank you for highlighting these inaccuracies. We have modified the manuscript checking the figure citations and the reference format.

Referee #3 (Remarks for Author):

The manuscript explores the mechanisms involved in a potential oncogenic function for mir9 in HNSCC. This study builds on a previous study from this group (PMCID: PMC7309652), where they identified a four-miRNA signature that classified HNSCC patients at high- or low-risk of recurrence. They showed that miR-9, targets SASH1 and KRT13 (confirmed here), whereas miR-1, miR-133, and miR-150, collectively target SP1 and TGF β pathways. In this study the authors now claim that mir9 also regulates SP1. This suggests a complex regulation of SP1 by multiple miRNAs in HNSCC. The experiments show that mir9 can induce the expression of SP1, through KLF5 suppression, which may contribute to promote tumorigenesis in HNSCC cells. The manuscript provides large amounts of data, but there is a lack of consistency and clarity in the data presentation that makes the paper hard to read: (1) there is little flow on the data description; (2) some experiments seem to contradict each other or just not support claims from previous figures; (3) it is unclear why they switch cell lines in some experiments; (4) in some experiments they show RNA expression of mir9 targets, in others they show protein, with no apparent reason; (5) different loading controls are used in different westerns, sometimes only ponceau staining is shown; (6) some comparative analyses might be statistically significant but the differences between groups are small and it is hard to predict whether those differences are biologically relevant which might question the significance of these studies; (7) the chemoradiation experiments (Fig 5) add little to the mechanistic studies of mir9 activities, and in fact, they add confusion because the conclusions are not well supported. (8) quantification of western blots and some staining is generally lacking.

We thank the Reviewer for highlighting strength and weaknesses of our manuscript. We believe that following Her/His suggestions the manuscript has been significantly improved and that the data are now clearer and also more robust. We really hope that in the present form She/He will find the manuscript acceptable for publication in EMBO Molecular Medicine.

Below are specific comments:

1. They use two cell lines (Fadu and Cal27) in most experiments, but in some experiments one of these cell lines is substituted by SCC9. In most experiments they use one cell line for gain-of-function studies and another one for loss-of-function. They should use at least 2 cell lines for each gain-of-function and loss-of-function. SCC9 should have been included in the initial characterization of the cell lines shown in EV1.

We agree with the Reviewer on this point. To properly address this question, we have now completely characterized two cell lines (FaDu and CAL27) and confirmed the most relevant data on

two other models, one for loss-of-function (SCC9) and one for gain-of-function (UMSCC1) experiments, now reported in Fig EV4, and Appendix Figures S2, S3, S5 and S6.

2. The criteria used to count colonies in clonogenic assays should be described. In some experiments (e.g. Fig. EV4 C) the graphs don't seem to match the images shown.

We apologize for this inaccuracy. We now provide a better description of this assay in the Methods section of Supplementary Information, for space limitation.

Briefly, before performing the IR assay, we analyzed the clonogenic ability of miR-9 modified cells. Given the strong effect of miR-9 in mediating an increased cell survival, we needed to differently plate shCTR or miR-9 cells, just to obtain a similar number of colonies in the untreated condition.

This formula is commonly used to calculate the correct number of cells to be plated before irradiation (e.g. we used this method in PMID 28377607).

$$N^{\circ} \text{ cell} = N^{\circ} \text{ optimal counting colonies/plating efficiency in standard conditions/likelihood of predicted survival.}$$

We then calculated the survival fraction, as follows:

$$\text{Survival fraction} = N^{\circ} \text{ clones in the IR condition} / N^{\circ} \text{ clones in untreated condition.}$$

3. Page 5 "In a panel of HNSCC-derived cell lines". Two cell lines cannot be considered a panel.

As mentioned in the text with this expression we principally referred to the previous manuscript, cited as reference, in which we analyzed 7 HNSCC derived cell lines for miR-9 expression (Figure S1A in Citron et al. 2017). However, we have now included in our analyses also SCC9 and UMSCC1 cells. In this small panel we also included normal squamous epithelial cells (NHBE) (see new Appendix Fig S1A) and confirmed our previous observation.

4. Page 5: "FaDu cells, associated with a mesenchymal phenotype". Mesenchymal markers are not shown. Same comment for the overexpression of mir9 in CAL27 (Fig EV2).

We agree with the Reviewer and we now rephrased the sentence focusing on epithelial markers (see new Fig 1B, Fig EV1B and Appendix Fig S2B, S2G and S2K).

5. Mir9 expression is shown in two HNSCC cell lines, but not in normal oral epithelial cells. This is important to see the extent of upregulation of mir9 in these two cell lines.

We have now analyzed miR-9 expression by qRT-PCR in all the HNSCC cell lines and compared it to normal bronchial epithelial cells (NHBE), as normal control, since we were not able to find

normal epithelial cells from head and neck organs. These results are also confirmed by literature data (e.g. PMID: 22133638 and 27694005), demonstrating that miR-9 is upregulated in HNSCC tumors compared to the normal tissue.

6. Fig. 1. mir9 suppression leads to less than 50% reduction in mir9 expression. I am not sure if this is biologically significant. This is reflected in a moderate change in the mir9 targets KRT13 and SASH1. This comment affects many of the comparative analysis between controls and mir9 gain- or loss-of-function cells.

We thank the Reviewer for giving us the possibility to explain better and more in detail these results. We analyzed microRNA expression by qRT-PCR. However, this method is not the best to evaluate miR expression when anti-miRs are used (e.g. reviewed in PMID: 22230293). The high levels of antimiR in the RNA sample may interfere with the detection step of the assay, for instance during the primer annealing or extension steps in miRNA-specific real-time qPCR.

For this reason, orthogonal approaches are strongly suggested in order to corroborate and validate the analysis. In general, the direct measurement of putative microRNA targets by Western Blot analysis is highly recommended. Also, it is strongly suggested to include functional effects after miRNA antagonism in the cell lines.

This is why we provided analyses of confirmed miR-9 targets in HNSCC and repeated the experiments in different models. We believe that all collected and presented data support the biological significance of our results.

7. The expression of the proteins shown in Fig 1B needs to be quantified, after normalization to loading control.

As requested, we have now included the quantification of mean intensity of SASH1, KRT13 and ZO-1 analyzed in three biological replicates (see new Fig. 1B). For the purpose of our manuscript, we decided to quantify SASH and KRT13, as a *bona fide* target of miR-9 (PMID: 28174235) and ZO-1 as a surrogate of pro-epithelial phenotype.

To complete the analysis, in the new version of the manuscript we also provide the expression and quantification of SASH1, KRT13 and ZO-1 in all the cell lines modified for miR-9 expression in the manuscript (see new Fig EV1B, and Supplementary Fig S2B, S2G and S2K)

8. Fig 1G. The data shown in this figure is contradicted by that shown in Fig 6A (untreated)

We partially disagree with the Reviewer on this point. In Fig. 1G-H, we injected FaDu cells stably transduced with shCTR or antimiR-9 vectors. In this experiment, we observed that inhibition of

miR-9 strongly delayed the tumor onset. This phenotype could partially mask synergistic effects of the treatments. For this reason, in the experiment Fig. 6A, we decided to inject FaDu parental cells in all the animals, thus abolishing all differences in terms of proliferation/onset due to miR-9 inhibition. Our aim was to understand whether the inhibition of miR-9 could have a synergistic effect in combination with IR and IR+CTX. Thus, once the tumors were fully engrafted and reached a $\sim 50 \text{ mm}^3$ of volume, we used intra-tumors injections of high-titer lentiviral particles encoding for shCTR or antimiR-9 and then proceeded with treatments, as represented in Fig. 6A. Altogether, results collected are not contradictory, as two very different approaches were used, but, conversely, are in line with the tumor take rate experiment (now shown in the new Fig 1G and Fig EV1G) and with the self-renewal experiments (Figure 1 and Fig EV1). The fact that, in the experiment Fig 6A-B, we did not observe any difference between the shCTR and antimiR-9 cells in the untreated condition, in our opinion, is due to the fact that miR-9 seems to play a pivotal role in the early steps of tumor engrafting and in the response to therapies (IR and/or CTX), while much less (if any) in the tumor growth.

9. Fig 2. Western blot for EGFR phosphorylation in Cal27 cells with silenced mir9 should be shown, as nicely shown for mir9 overexpression in Fig 2G.

We thank the Reviewer for raising this point. We reasoned that to make our manuscript easier to read and to better describe to most salient parts of our work was better to not investigate further the activation of EGFR in response to different levels of miR-9, focusing only on the up-regulation of miR-9 in response to mitotic stimuli, including EGF. We therefore removed this experiment from the manuscript.

10. Fig 2H tries to make the point that a correlation between mir9 and EGFR exists in HNSCCs. However, the data indicates that this correlation is very weak ($r=0.18$).

We agree with the reviewer on this point. Yet very recent data on HPV negative HNSCC demonstrate that in these tumors the absolute levels of EGFR and/or its amplification failed to predict the activation of the pathway and/or the response to CTX (PMID 33417831). In light of these new data, we believe that a weak correlation could be expected and could at least partially support our *in vitro* and *in vivo* data. We better discuss this point in the discussion section (see page 17 lane 14-18).

11. The resistance to Gefitinib shown in Appendix Figure S1 is very modest for SCC9. I wonder why Cal27 was not used in these experiments to confirm the more robust effects observed in Fadu.

We agree with the Reviewer on this point and we have now broadened our analyses. In all tested model, the resistance to Gefitinib in cells expressing miR-9 is weak (except for FaDu cells). From a translational standpoint, it is worth to underline that CTX, and not Gefitinib, improved overall survival when given with radiotherapy in advanced HNSCC patients (e.g. PMID: 21821303) and this is the reason why we used both Gefitinib and CTX in *in vitro* experiments.

We thus tested CTX sensitivity in UMSCC1 and CAL27 cells expressing or not miR-9, now shown in the new Appendix Fig S3B-C. We observed that all the cell lines expressing high level of miR-9 showed a marked resistance to the monoclonal antibody.

12. Gefitinib induces a rapid downregulation of mir9 on Fadu (Fig. 2C), as 2h after treatment approximately 75% of the mir9 expression is lost, a greater decrease in mir9 than that obtained with the anti-mir9 (Fig 1A). However, the cell viability is only affected when mir9 is suppressed with anti-mir9. This should be discussed and confirmed with other cell lines.

As we have discussed at point 6 above, it is difficult to measure the inhibition of miR-9 in anti-miR-9 cells. In order to address this question, we included in our analysis CAL27 cells expressing or not miR-9 and treated with increasing doses of Gefitinib (Fig EV2D). It is interesting to note that in this case CAL27 cells did not display a frank resistance to Gefitinib, even when miR-9 was expressed.

13. Fig. 3A lacks units.

We have now corrected the graph.

14. Fig. 3C shCRT-control and anti-mir9-Control show about the same expression of SP1. According to Fig. 3B, should SP1 expression be lower in mir9-Control?

We thank the Reviewer for this observation. As shown in the graph of new Fig. 3B, the expression of Sp1 is significantly higher in shCTR respect to anti-miR-9 FaDu cells. We provide the statistics in the new figure, which was not reported before.

15. Fig. 3E-F. SP1 is known to control cell growth. The effects on SP1 suppression in cell growth in HNSCC cell lines should be shown. If cell growth is compromised by SP1 suppression, how would that affect the clonogenic and sphere formation ability of the cells?

We agree with the Reviewer on this point and we better characterized the role of SP1 on cell proliferation and sphere forming assay, both in FaDu and CAL27 cells, silenced or not for SP1 (see new Fig 3E-H, EV2D and Appendix Fig S5). As expected, SP1 silencing significantly reduced cell proliferation and sphere forming efficiency in both cell lines, confirming that it could mediate the

effects of miR-9. This possibility was confirmed looking at the effects of SP1 in the response to Gefitinib and CTX in control and antimiR-9 FaDu and CAL27 cells, showing that SP1 overexpression reverts the high sensitivity of antimiR-9 cells to EGFR blockade.

16. The concentration of Mithramycin A used in different experiments should be indicated.

We have now added the concentration of Mithramycin A used in the different experiments in the new Fig. S5E-F.

17. What is the control in the luciferase assays?

In the luciferase assay Fig. 2A and 2C, the control is represented by FaDu parental cells transfected with the pBabe miR-9 sponge vector and treated with vehicle. Data are expressed as normalized luciferase activity, folded on to the untreated condition.

In the luciferase assay testing the SP1 promoter activity (new Fig. 3J and EV2E) the control is represented by FaDu cells transfected with pGL3 empty vector. Data are expressed as normalized luciferase activity folded on to the empty vector.

18. Fig. EV3F-L are not referenced correctly in the text.

We apologize for this mistake. We have now corrected the manuscript.

19. Fig. EV3H. How was the 143-genes signature from Pavon et al generated? and why was it used to compare with the mir9 targets?

We apologize for not explaining this point clearly in the paper. The methodology that we followed to generate the “Pavon’s signature” was described in Citron et al 2017. Briefly, based on the knowledge that miR-9 is associated with mesenchymal features and de-differentiation processes in different tumors types, including HNSCC, we interrogated the expression profile of 63 treatment-naïve tumor biopsies from advanced HNSCC patients included in Pavon et al (PMID 22696598), that described 191 genes up-regulated in tumors from patients that did not had recurrences and down-regulated in tumors from recurrent patients. Then, we intersected these genes with the 1611 potential miR-9 targets. This approach led to the identification of 20 genes, listed in Appendix Fig S3A.

20. Fig. 4C. KLF5 induces, at the most, a modest reduction in SP1 expression. SP1 expression should be quantified after normalization to loading control. GRB2 is an unusual loading control.

We have now provided a new WB analysis along with the quantification graph for Sp1 in FaDu cells transfected with KLF5 or empty vector (New Fig. 4F), as well as, the quantification graph for both Sp1 and KLF5 in FaDu cells expressing or not miR-9 (New Fig. 4B). We used also other loading controls although GRB2 is one of our favorite ones (e.g. Citron et al. 2017). These new analyses clearly show that high expression of KLF5 results in low expression of Sp1.

21. Fig. 4E. The Chip'd DNA for control IgG is not shown.

We thank the Reviewer for this comment. Now, we have better explained how the former Fig. 4G was assembled to include all relevant controls (new Fig 4H). We show the WB analysis highlighting the specificity of the antibody used for the IP of KLF5 in ChIP and an agarose gel showing that all primers designed for the amplification of Sp1 promoter in the ChIP analyses specifically amplified both the DNA eluted from the ChIP assay and the genomic DNA of FaDu cells (no smears and a single band present on the gel after 40 cycles of PCR).

We have now graphed the ChIP data as signal relative to input and we also have included new ChIP analyses performed in CAL27 cells (new Fig 4H and Fig EV3G).

The panels below report the ChIP experiments performed in FaDu (A) and CAL27 (B) cells.

Negative (IgG) and the positive (Histone H3, provided in the commercial kit) controls are included and the data are presented as % of input. We have also calculated the folds over control (IgG) and the results are extremely comparable (see panel C and D of the figure below, not included in the manuscript). The new Method section reports in detail how ChIP was performed and how % of input was calculated. Statistical significance is reported and calculated with a two-way ANOVA test.

In the manuscript we have omitted to include the Histone H3 ChIP that, in our opinion, represents a redundant control. However, if the Editor and/or the Reviewer think that it would be necessary to have this control in the manuscript we are willing to add it.

22. Fig. 5. Drug/radiation response was analyzed at 72h. Considering that mir9 suppression results in slow cell growth, the response to the drug and radiation might be partially masked the cell growth effects. It would be cleaner to analyze the drug/radiation response at 24h. Cell density also affects the response. According to Fig 5D cell density of anti-mir9 cells was lower than that of control cells.

We thank the Reviewer for this comment. We have now analyzed the drug sensitivity (CDDP, TAX, 5-FU, BLEO, GEFI and CTX) after 24hrs of treatment in different cell lines, modified for miR-9 expression (CAL27, FaDu and SCC9). As reported below, we observed significant differences in drug sensitivity only when CDDP and TAX were tested, even though these differences cannot be attributed to miR-9 expression (panel A and B). We did not observe any changes in drug sensitivity when 5-FU was tested (panel C). Finally, we confirmed that miR-9 expression correlates with resistance to radio-mimetic agent bleomycin (BLEO, panel D), in CAL27 and SCC9 cells. Further, miR-9 inhibition appeared to mediate sensitivity to EGFR blockade in both FaDu and SCC9 cells treated with GEFI or CTX, further supporting the importance of miR-9 in mediating this phenotype. Since these data do not alter the results of our work, we provide them as information for the Reviewer only. However, if the Editor or the Reviewer think it is necessary to include the data in the main manuscript, we are willing to do so.

In the figure below, data are expressed as mean value +/- SD, and statistical significance was calculated with Student t test. * $p < 0.05$, ** $p < 0.001$, < $p < 0.0001$.

23. Fig 5C. According to the images of the cell dishes, mir9 overexpression doesn't seem to affect significantly the response to radiation. It seems that most of the effects conferred by mir9 are on cell growth.

We have now described (point 2) the method we have used for calculating the number of cells in the radiation assay. This calculation is meant to plate cells differently, based on their different

ability to grow and/or to survive in colony formation assay, in order to see the different response to radiation with no other confounding effects. Moreover, our results deriving from orthogonal approaches (survival fraction, kill curves, WBs, IFs, *in vivo* experiment) are highly consistent, showing that miR-9 inhibition sensitized HNSCC cells to radiation-induced death.

24. Fig EV4D. Quantification should be shown.

25. Fig EV4E. Difference in γ H2AX between control and mir9 is really small, might not be biologically significant. For pS10H3 it's concluded that radiation induces cell cycle arrest in control, but not in mir9 cells? What is the interpretation of the pS10H3 data in Fig 5D?

We thank the Reviewer for the suggestion. We have decided to remove the IF analysis performed in SCC9, because we added new analysis in FaDu, SCC9, CAL27 and UMSCC1 modified for miR-9 expression and irradiated with 2 or 5Gy. As shown by WB and IF analyses (new Fig 5C-D, Fig EV4C and Appendix Fig S6C-D), anti-miR-9 FaDu and SCC9 cells displayed an increased and persistent activation of H2AX, marker of DNA damage, compared to controls. On the other side, miR-9 overexpression in CAL27 and UMSCC1 cells led to a decreased level of γ H2AX.

26. Fig 5E. The protein expression needs to be quantified, after normalization.

27. Fig 5E-F: "in irradiated FaDu cells, SP1 expression paralleled miR-9 levels showing a transient reduction of both protein and mRNA levels (Fig 5E-F)". This transient reduction of SP1 needs to be supported by the data; the protein was not quantified. And the reduction in RNA levels seems to persist at all time points (Fig. 5F).

We have now provided a new WB analysis, performed on FaDu and SCC9 silenced or not for miR-9 expression and irradiated with 5 and 2 Gy, respectively (Fig 5C and Fig EV4C). The activation of γ H2AX is clearly higher in both cell lines upon miR-9 loss, yet we agree to include the quantification of Sp1 expression. The data confirmed our previous results, showing that anti-miR-9 FaDu and SCC9 cells express a lower level of Sp1, compared to controls (Fig 5C and Fig EV4C), at all time points after radiation.

28. Fig 5F: "Moreover, in anti-miR-9 FaDu cells SP1 was significantly downregulated respect to controls and its expression did not recover up to 24 hours after irradiation (Fig 5E-F)". According to the data, it seems that radiation downregulates SP1 RNA in control but not in anti-mir9 cells.

We thank the Reviewer for this suggestion. We have rephrased our description, according to the data shown (page 12, lines 17-19).

29. Fig EV4F. Protein needs to be quantified.

We thank the Reviewer for this suggestion. As specified above, we have now provided a new WB as well as the Sp1 quantification, analyzed in FaDu and SCC9 silenced or not for miR-9 expression and irradiated with 5 and 2 Gy (Fig 5C and Fig EV4C).

30. Fig 6A. Fadu cells seem to be resistant to EGFR. In addition to validate this experiment with at least another cell line, EGFR expression and activation should be analyzed in control and anti-mir9 cells.

As requested, we analyzed the response to EGFR blockade, using Cetuximab in FaDu, SCC9 and CAL27 cells silenced for miR-9 expression. As discussed previously, when miR-9 expression was inhibited in SCC9 and FaDu cells, we observed a significant increase in CTX and Gefitinib sensitivity respect to controls (new Fig 2G-H and Appendix Fig S3A). With regards to CAL27 cells, displaying low endogenous level of miR-9, miR-9 inhibition did not further affect their sensitivity to CTX treatment (Fig EV2D). Of note, the restoration of Sp1, by forced overexpression, completely reverted the phenotype, strongly suggesting that the axis miR-9/Sp1 is a mechanism leading to CTX resistance in HNSCC (Fig 3G-H and Fig EV2D). We also analyzed the expression of pY1068 EGFR and total EGFR in CAL27 shCTR or anti-miR-9 cells, expressing or not Sp1. As shown in Fig EV2C, in exponential growth condition, there is no difference in the activation and expression of EGFR.

31. Page 11. "Interestingly, SP1 expression was reduced in tumors formed by anti-miR-9 FaDu cells (in 8/10 mice)". This statement is not supported by the data; in the 10 pairs of tumors (control and anti-mir9), SP1 expression is lower in anti-miR-9 in 4 of them, and higher or unchanged in 6 of them. The quantification graph shown in Fig EV5C does not seem to match the intensity of the SP1 bands for mice #2 and #4, even after normalization of GAPDH.

We thank the Reviewer for raising this point. To better address Her/His concern, we have now used immunohistochemistry (IHC) to precisely evaluate Sp1 expression in tumor cells (new Fig. EV5D). Sp1 expression was graded by a blinded pathologist based on nuclear staining intensity (0=negative, 1=very weak, 2=weak, 3=moderate, 4=strong). Collected data demonstrate a significant reduction in Sp1 expression in miR-9 silenced cells.

32. Fig 6C. The response to the combination in control and anti-mir9 cells might be statistically significant, but the difference is so small that is hard to see that this data is biologically significant.

In order to clarify this point, we include below the graphs separated for treatment to better highlight the biological differences in our experiment. As shown in graph A (IR group) and in graph B (IR + CTX), the volume of antimiR-9 tumors was in both cases approximately the half respect to the controls and the tumors from antimiR-9 cells start to regrow later than those from control cells.

33. Fig 6D needs to be quantified.

We apologize with the Reviewer for the lack of clarity. The quantification of Ki67 expression was already included in Fig. EV5F. Now we have moved the graph close to the IHC staining for Ki67, in new Fig. 6D-E.

34. The number of patients in each group should indicated (Fig6 E-F, Fig EV5I).

We thank the Reviewer for the suggestion and we have now included the number of patients in each group.

35. Does CTX treatment in mice affect the mir9 expression in tumors?

We thank the Reviewer for this question. To address this point, we analyzed miR-9 expression by qRT-PCR in tumors arising from FaDu shCTR cells and treated them with vehicle or CTX, as described in Fig EV5A. We observed that CTX administration did not affect miR-9 expression respect to untreated animals, as expected since from our data the administration of CTX alone in nude mice did not exert the optimal effect. We provide this result as information for the Reviewers. However, if the Editor and/or the Reviewer think it is necessary to include the data in the main manuscript, we will be willing to do so.

Dear Dr. Baldassarre,

Thank you for the submission of your revised manuscript to EMBO Molecular Medicine. We have now heard back from the two referees who we asked to re-evaluate your manuscript. As you will see from the reports below, the referees are overall supporting publication of your manuscript but also raise some concerns that should be addressed in an additional and final round of revision. Please implement all adjustments suggested by the referees. Addressing point 4 and 5 from referee #2 is essential and will entail additional round of review. Acceptance or rejection of the manuscript will depend on the completeness of your responses included in the next, final version of the manuscript. For this reason, and to save you from any frustrations in the end, I would strongly advise against returning an incomplete revision.

In addition, please amend the following:

1) In the main manuscript file, please do the following:

- Provide up to 5 keywords.
- Rename Supplementary Table S1 and S3 (page 21) to Appendix Table S1 and S2.
- Legend for Fig EV 5I should be corrected to EV 5G.
- Callout for Appendix Fig EV 5G (page 16) should be corrected to Fig EV 5G.
- Rename "Disclosure of Potential Conflicts of Interest" to "Conflict of interest".
- Add contribution for Vittorio Giacomarra. Please use author's initials instead of full last name. Make sure that authors with same initials can be distinguished.
- In M&M, provide the antibody dilutions that were used for each antibody.
- In M&M, include a statement that informed consent was obtained from all human subjects and that the experiments conformed to the principles set out in the WMA Declaration of Helsinki and the Department of Health and Human Services Belmont Report.
- In M&M, the statistical paragraph should reflect all information that you have filled in the Authors Checklist, especially regarding randomization, blinding, replication.
- Correct the reference citation in the reference list. Citations should be listed in alphabetical order. Where there are more than 10 authors on a paper, 10 will be listed, followed by "et al.". Please check "Author Guidelines" for more information.

<https://www.embopress.org/page/journal/17574684/authorguide#referencesformat>

- Add data availability statement. If no data are deposited in public repositories, please add the sentence: "This study includes no data deposited in external repositories". Please check "Author Guidelines" for more information.

<https://www.embopress.org/page/journal/17574684/authorguide#availabilityofpublishedmaterial>

2) Appendix:

- Move supplementary Materials and Methods to the main manuscript file.
- Leave primer tables and name them Appendix Table S3 and S4 with the title and appropriate callouts in the manuscript text.

3) Funding: CRO-Aviano 5% grant for G.B is missing in our submission system. Please make sure that information about all sources of funding are complete in both our submission system and in the manuscript.

4) The paper explained: Please shorten this section to better emphasize the major findings of the paper and their medical implications for the non-specialist reader. Please refer to any of our

published primary research articles for an example. As suggested by the referee #1 please also revise the text for grammar and syntax (i.e. by an English native speaker).

5) For more information: This section should include relevant web links for further consultation by our readers. Could you identify some relevant ones and provide such information as well? Some examples are patient associations, relevant databases, OMIM/proteins/genes links, author's websites, etc...

6) Synopsis:

- Please redesign and/or resize the synopsis image to 550 px-wide x (250-400)-px high jpeg image.
- Please remove synopsis text from the main manuscript and submit it as a separate .doc file. As suggested by the referee #1 please also revise the text for grammar and syntax (i.e. by an English native speaker).

7) As part of the EMBO Publications transparent editorial process initiative (see our Editorial at <http://embomolmed.embopress.org/content/2/9/329>), EMBO Molecular Medicine will publish online a Review Process File (RPF) to accompany accepted manuscripts. This file will be published in conjunction with your paper and will include the anonymous referee reports, your point-by-point response and all pertinent correspondence relating to the manuscript. Let us know whether you agree with the publication of the RPF and as here, if you want to remove or not any figures from it prior to publication. Please note that the Authors checklist will be published at the end of the RPF.

8) Please provide a point-by-point letter INCLUDING my comments as well as the reviewer's reports and your detailed responses (as Word file).

Please submit your revised manuscript within three months. I look forward to reading a new revised version of your manuscript.

Yours sincerely,

Zeljko Durdevic

***** Reviewer's comments *****

Referee #1 (Comments on Novelty/Model System for Author):

Extensive evaluation in various cell lines and in model systems.

Referee #1 (Remarks for Author):

Minor odd wording on line 10 in the abstract "... expression correlates with the one of SP1 and high" Same statement on line 30 of page 4.

The section "The paper explained" and "Synopsis" need basic editing for English and grammar.

Introduction, line 6 needs editing for clarity "... tumors in 50% of patients eventually recur..." Tumors recur, patients do not.

More methods are needed regarding transwell assays (Appendix F4J). Why are cells allowed to migrate and then "reattach" for 8 hours? This is not common. Imaging of cells that have crossed through the Matrigel and are on the underside of the chamber should be imaged and used as data for the migration assays. The images also are lacking the typical transwell pores. The description in the text does not match the methods which state that post migration fluorescence was obtained from the top or bottom of the chamber. How were cells fluorescing? This is also lacking in description. Image shown is white-light, not fluorescence.

I want to draw attention to EV1B where the authors show that overexpressing miR-9 downregulated a few of the previously reported miR-9 target genes. The western blot indicates substantial down regulation of these targets; however, the quantification does not appear to accurately reflect this downregulation. Not only are the data completely opposite - the miR-9 group in the bar graph indicates increased protein, but the difference between the two treatment groups does not appear to reflect the strong difference shown in the western blot. This data should be reassessed by the authors.

Authors should reassess data in figure 2B and adjust the text accordingly. The text states that EGF increased miR-9 expression 3-4 fold, which is inaccurate based on the data presented. The increase in miR-9 is at most 3 fold, but more accurately, 1.5-3 fold, opposed to 3-4 fold.

For the ChIP data presented in Figure 4H, it would be beneficial to direct the reader, to the legend of the previous schematic of the promoter of SP1 indicating the various primer pairs used for amplification. Basically, indicating that putative SP1 binding sites and primer locations are depicted in Figure 3I.

Referee #2 (Remarks for Author):

The authors have addressed most of my concerns, however few points still need clarification before publication:

1. The mutations of TP53 in Fadu and Cal27 cells are missense ones, these should not change the molecular weight of the protein. In my opinion, a comment on the different size of the bands detected by WB should be included in the text. In my opinion, the table indicating the mutations carried by the different cell lines will facilitate the readers.

2. The authors should explain why the mesenchymal marker N-Cadherin has been deleted from replicates of WB analysis (fig. 1B).
3. The sentence "when both the seed sites in 3'UTR of KLF5 were mutated (mut A+B), miR-9 modification failed to modulate KLF5-driven LUC activity" is not entirely correct since the mut A+B (fig 4 C) is still sensible to the antimir-9 treatment in FaDu cells (Fig 4D). Please fix this point.
4. There was a misunderstanding concerning the positive and negative controls to be included in ChiP experiments. I meant internal controls; for instance, an intergenic genomic region which is usually used in RNA-Pol II ChiP can be employed as negative control while public data can be used to find a positive control (DOI: 10.1158/0008-5472.CAN-20-1287; <https://doi.org/10.1101/2020.02.10.941872>). At least one of these controls is essential to verify the specificity of the KLF5 binding on SP1 promoter. Histone H3 ab is provided by the commercial kit just to set the technique and not as positive control for specificity of the used ab. Moreover, the authors stated that they performed two independent experiments, however, there is only one IgG control in the graph. It is also hard to understand how they applied the two-way ANOVA test.
5. Even though the authors extended the WB analysis of 5Gy IR treated cells to other cell lines (new Fig 5C, EV4C and Appendix S6C, this latter lacking quantification) still, each experiment is performed once. The quantification on one gel cannot allow to derive the conclusion: "A rapid and more sustained expression of H2AX was observed in all tested cell lines when miR-9 expression was lower than controls". Provide replicates, quantification and statistic at least for the most representative cell lines.

The authors performed the requested editorial changes.

Reviewer's comments:

Referee #1 (Comments on Novelty/Model System for Author):

Extensive evaluation in various cell lines and in model systems.

We thank Reviewer#1 for Her/His appreciation of our work and we hope that She/He will find our work acceptable for publication in EMBO Molecular Medicine.

Minor odd wording on line 10 in the abstract "... expression correlates with the one of SP1 and high" Same statement on line 30 of page 4.

We thank Reviewer#1 for highlighting this point and we have now modified accordingly. Please, see new abstract line 9-11 page 3 and new synopsis.

The section "The paper explained" and "Synopsis" need basic editing for English and grammar.

We have thoroughly edited all manuscript for English and grammar.

Introduction, line 6 needs editing for clarity "... tumors in 50% of patients eventually recur..."

Tumors recur, patients do not.

We have modified the text accordingly.

More methods are needed regarding transwell assays (Appendix F4J). Why are cells allowed to migrate and then "reattach" for 8 hours? This is not common. Imaging of cells that have crossed through the Matrigel and are on the underside of the chamber should be imaged and used as data for the migration assays. The images also are lacking the typical transwell pores. The description in the text does not match the methods which state that post migration fluorescence was obtained from the top or bottom of the chamber. How were cells fluorescing? This is also lacking in description. Image shown is white-light, not fluorescence.

We thank Reviwer#1 for this comment. We apologize for the lack of clarity and we have now provided a detailed description of the migration assay in the method section.

Briefly, as reported in the Material and Methods section, we quantified the PKH26 fluorescence signal from the bottom and the top of a transwell (carrying a FluoroBlok membrane) over a period of 2 hours, using a plate reader. After the quantification of migrated cells, we decided to allow the cells to re-attach to the plate just to take a representative picture of viable cells that efficiently migrate. This latter step represent a modification of published procedures that needs to disassemble the transwell to take picture of migrated cells from the bottom side of the membrane (e.g. PMID: 20194624; 21423803).

I want to draw attention to EV1B where the authors show that or expressing miR-9 downregualted a few of the previously reported miR-9 target genes. The western blot indicates substantial down regulation of these targets; however, the quantification does not appear to accurately reflect this downregulation. Not only are the data completely opposite - the miR-9 group in the bar graph indicates increased protein, but the difference between the two treatment groups does not appear to reflect the strong difference shown in the western blot. This data should be reassessed by the authors.

We thank Reviewer#1 for raising this point. We partially disagree with Her/Him on this point. The graph reported in Fig. EV1B is the result of three independent experiments, in which we quantified the miR-9 targets (Zo-1, SASH1 And KRT13) normalized over the Histone H3 expression. We believe that the graph well mirrored the WB analyses provided. Yet, we agree that the graph is difficult to interpret. For space limitation, due to the presence of multiple panels in the figure, we reported the expression of the three genes in miR-9 overexpressing cells (blue) and control cells (gray) on the same bar. We also provide here below the graph with the separated columns that readily demonstrate the observed differences.

Authors should reassess data in figure 2B and adjust the text accordingly. The text states that EGF increased miR-9 expression 3-4 fold, which is inaccurate based on the data presented. The increase in miR-9 is at most 3 fold, but more accurately, 1.5-3 fold, opposed to 3-4 fold. We apologize for this inaccuracy; we have now corrected.

For the ChIP data presented in Figure 4H, it would be beneficial to direct the reader, to the legend of the previous schematic of the promoter of SP1 indicating the various primer pairs used for amplification. Basically, indicating that putative SP1 binding sites and primer locations are depicted in Figure 3I.

We thank Reviewer#1 for this suggestion and we modified the legend to Figure 3I and 4H, accordingly.

Referee #2 (Remarks for Author):

The authors have addressed most of my concerns, however few points still need clarification before publication:

We thank Reviewer#2 for Her/His positive comments and we hope that She/He will now find our work acceptable for publication in EMBO Molecular Medicine.

1. The mutations of TP53 in FaDu and Cal27 cells are missense ones, these should not change the molecular weight of the protein. In my opinion, a comment on the different size of the bands detected by WB should be included in the text. In my opinion, the table indicating the mutations carried by the different cell lines will facilitate the readers.

We thank Reviewer#2 for this comment. Stimulated by Her/His request we better searched in all available databases and we found that FaDu cells carry two different mutations in the TP53 gene, as reported in the table below:

Gene Sequence	Protein Sequence
c.743G>T	p.R248L
c.376-1G>A	p.?

In particular, the mutation c.376-1G>A possibly affects the splicing form of p53 transcript, and this may result in a protein with a lower molecular weight, as shown in Appendix Fig S1B (see link below for more information).

[https://www.ncbi.nlm.nih.gov/clinvar/?term=%22MutSpliceDB%3A%20a%20database%20of%20splice%20sites%20variants%20effects%20on%20splicing%2CNIH%22\[submitter\]+AND+%22TP53%22\[gene\]](https://www.ncbi.nlm.nih.gov/clinvar/?term=%22MutSpliceDB%3A%20a%20database%20of%20splice%20sites%20variants%20effects%20on%20splicing%2CNIH%22[submitter]+AND+%22TP53%22[gene])

We decided to not include the table reporting the TP53 mutations of the different cell lines since they are retrieved from literature and are readily available in the ATCC website and DepMap portal for SCC9, FaDu and CAL27, and in Bradford CR, et al Head Neck 2003 (Ref. 21 in the manuscript) for UMSCC1.

However, also to meet the Editor request to add relevant web links in the section “For more Information”, for further consultation by EMM readers, we have added the links in which the mutational status of CAL27, FaDu, and SCC9 cells are reported.

<https://www.atcc.org/~media/C942A3363ED74FC0AAB46BE45D58ED1D.ashx>

https://depmap.org/portal/cell_line/ACH-000832?tab=mutation

2. The authors should explain why the mesenchymal marker N-Cadherin has been deleted from replicates of WB analysis (fig. 1B).

We removed the mesenchymal marker N-Cadherin because it is not expressed in all the cell lines tested in this manuscript, thus for consistency and to give a clearer message, we decided to analyze only ZO-1, as marker for the epithelial phenotype, and KRT13/SASH1, as miR-9 targets, described in our previous work Citron et al CCR 2017.

3. The sentence "when both the seed sites in 3'UTR of KLF5 were mutated (mut A+B), miR-9 modification failed to modulate KLF5-driven LUC activity" is not entirely correct since the mut A+B (fig 4 C) is still sensible to the antimir-9 treatment in FaDu cells (Fig 4D). Please fix this

point.

We have rephrased in the manuscript as suggested, see lines 9-11 page 13.

4. There was a misunderstanding concerning the positive and negative controls to be included in ChIP experiments. I meant internal controls; for instance, an intergenic genomic region which is usually used in RNA-Pol II ChIP can be employed as negative control while public data can be used to find a positive control (DOI: 10.1158/0008-5472.CAN-20-1287; <https://doi.org/10.1101/2020.02.10.941872>). At least one on these controls is essential to verify the specificity of the KLF5 binding on SP1 promoter. Histone H3 ab is provided by the commercial kit just to set the technique and not as positive control for specificity of the used ab. Moreover, the authors stated that they performed two independent experiments, however, there is only one IgG control in the graph. It is also hard to understand how they applied the two-way ANOVA test.

In the experiments, we used a ChIP grade anti-KLF5 antibody, we assumed that its specificity for this application was already guaranteed by the provider. Besides, the same Ab has been already used in many other published articles for ChIP assay experiments (e.g. PMID: 28440310; 30271790; 32332020). Moreover, “positive” and “negative” controls could be different depending on the cells that are analyzed, especially when cancer cells are tested. For instance, KLF5 binds the enhancer motif of HNF4A in gastrointestinal adenocarcinomas but not in squamous cell carcinomas (PMID: 32332020).

However, to meet Reviewer#2 request, we searched the literature for putative validated KLF5 target genes in FaDu or Cal27 cells, with scarce results. Therefore, we selected three hypothetically negative controls (Negative Control 1 PMID 32805052; Negative Control 2 PMID: 19593370; Negative Control 3: PMID: 20875108) and 6 possibly positive controls (EPPK1 PMID: 33827480; LAMC2, EPHA2, SERPINE1 and INPP4B PMID: 20875108; and SOX17 PMID: 24770696).

As reported in the new Figure EV3H, we observed no differences between IgG and KLF5 Ab in the three negative controls and in two hypothetically positive controls (*i.e.* EPPK1 and EPHA2). We confirmed again that KLF5 binds SP1 promoter (Fragment -673/-486) at the same extent of LAMC2 and slightly less than SOX17, SERPINE1 and INPP4B. Based on these results we can conclude that the KLF5 Ab we used was effectively ChIP grade.

Regarding the second request, the control IgG represents the mean of two almost identical controls ChIP. Below, for your reference, we display the graph with the two separate control IgG.

We agree with Reviewer#2 that the ANOVA test did not apply and we removed it.

5. Even though the authors extended the WB analysis of 5Gy IR treated cells to other cell lines (new Fig 5C, EV4C and Appendix S6C, this latter lacking quantification) still, each experiment is performed once. The quantification on one gel cannot allow to derive the conclusion: "A rapid and more sustained expression of H2AX was observed in all tested cell lines when miR-9 expression was lower than controls". Provide replicates, quantification and statistic at least for the most representative cell lines.

As requested, we have now provided the quantification of Sp1 and γ H2AX expression in biological triplicates of lysates from FaDu and SCC9 cells, modified for miR-9 and irradiated or not (see new Fig 5C and Fig EV6C).

We are pleased to inform you that your manuscript is accepted for publication and is now being sent to our publisher to be included in the next available issue of EMBO Molecular Medicine.

YOU MUST COMPLETE ALL CELLS WITH A PINK BACKGROUND ↓
PLEASE NOTE THAT THIS CHECKLIST WILL BE PUBLISHED ALONGSIDE YOUR PAPER

Corresponding Author Name: Gustavo Baldassare

Manuscript Number: EMM-2020-12872